# Single-cell transcriptomic evidence for dense intracortical neuropeptide networks

Stephen J Smith[1]*, Uygar Sümbül[1], Lucas T Graybuck[1], Forrest Collman[1], Sharmishtaa Seshamani[1], Rohan Gala[1], Olga Gliko[1], Leila Elabbady[1], Jeremy A Miller[1], Trygve E Bakken[1], Jean Rossier[2], Zizhen Yao[1], Ed Lein[1], Hongkui Zeng[1], Bosiljka Tasic[1], Michael Hawrylycz[1]*

[1]Allen Institute for Brain Science, Seattle, United States; [2]Neuroscience Paris Seine, Sorbonne Université, Paris, France

**Abstract** Seeking new insights into the homeostasis, modulation and plasticity of cortical synaptic networks, we have analyzed results from a single-cell RNA-seq study of 22,439 mouse neocortical neurons. Our analysis exposes transcriptomic evidence for dozens of molecularly distinct neuropeptidergic modulatory networks that directly interconnect all cortical neurons. This evidence begins with a discovery that transcripts of one or more neuropeptide precursor (NPP) and one or more neuropeptide-selective G-protein-coupled receptor (NP-GPCR) genes are highly abundant in all, or very nearly all, cortical neurons. Individual neurons express diverse subsets of NP signaling genes from palettes encoding 18 NPPs and 29 NP-GPCRs. These 47 genes comprise 37 cognate NPP/NP-GPCR pairs, implying the likelihood of local neuropeptide signaling. Here, we use neuron-type-specific patterns of NP gene expression to offer specific, testable predictions regarding 37 peptidergic neuromodulatory networks that may play prominent roles in cortical homeostasis and plasticity.

**\*For correspondence:**
StephenS@alleninstitute.org (SJS);
MikeH@alleninstitute.org (MH)

**Competing interests:** The authors declare that no competing interests exist.

## Introduction

Neuromodulation - the graded and relatively slow adjustment of fast synapse and ion channel function via diffusible cell-cell signaling molecules - is a fundamental requirement for adaptive nervous system function (*Abbott and Regehr, 2004*; *Bargmann, 2012*; *Bucher and Marder, 2013*; *Katz and Lillvis, 2014*; *Marder, 2012*; *Marder et al., 2015*; *McCormick and Nusbaum, 2014*; *Nadim and Bucher, 2014*; *Nusbaum et al., 2017*). Neuromodulator molecules take many different chemical forms, including diatomic gases such as nitric oxide, lipid metabolites such as the endocannabinoids, and amino acids and their metabolites such as glutamate, GABA, acetylcholine, serotonin and dopamine. By far the largest family of neuromodulator molecules known, however, comprises the evolutionarily ancient proteinaceous signaling molecules known as neuropeptides (*Baraban and Tallent, 2004*; *Burbach, 2011*; *Gonzalez-Suarez and Nitabach, 2018*; *Hökfelt et al., 2013*; *van den Pol, 2012*; *Wang et al., 2015*). The most widely studied neuropeptides are the endogenous 'opioid' peptides - enkephalins, endorphins and dynorphins - but there are nearly one hundred other NPP genes in the human genome and numerous homologs are present in almost all known animal genomes (*Elphick et al., 2018*; *Jékely, 2013*).

The broadest definition of 'neuropeptide' would embrace any soluble peptide that serves as a messenger by diffusing from one neuron to another. A narrower but more common definition (*Burbach, 2011*) requires (1) translation of a neuropeptide precursor protein (NPP) into the lumen of a source neuron's rough endoplasmic reticulum (rER), (2) enzymatic cleavage of the NPP into one or more neuropeptide (NP) products during or after passage through the rER–Golgi complex and

packaging into dense-core vesicles, (3) transport and storage of dense-core vesicles within the source neuron, (4) secretion of the NP product upon demand by activity- and calcium-dependent dense-core vesicle exocytosis, and then (5) interstitial diffusion to act upon a target neuron by binding to a ligand-specific receptor. This pathway enlarges the potential palette of distinct neuropeptides beyond that established simply by the large number of NPP genes, as a given NPP may be differentially cleaved into alternative NP products during its intracellular and interstitial passage.

Most neuropeptide receptors are encoded by members of the very large superfamily of G-protein-coupled receptor (GPCR) genes (*Hoyer and Bartfai, 2012*; *Krishnan and Schiöth, 2015*; *Mains and Eipper, 2006*; *van den Pol, 2012*). GPCRs are selective, high-affinity receptors distinguished by characteristic seven-transmembrane-segment atomic structures and signal transduction involving heterotrimeric G-proteins (hence their name). Genes encoding GPCRs selective for neuropeptides (NP-GPCR genes) and for most of the other chemical neuromodulators mentioned above are found in the genomes of almost all metazoans: phylogenomic evidence suggests very early evolutionary origins (*Jékely, 2013*; *Katz and Lillvis, 2014*), possibly even predating evolution of the synapse (*Varoqueaux and Fasshauer, 2017*). In modern metazoan nervous systems, synapses rely primarily upon recycling small molecule transmitters and ligand-gated ion channels (alternatively known as 'ionotropic receptors') for fast (millisecond timescale) transmission, but GPCRs selective for widely varied ligands including the fast recycling transmitters and many other secreted molecules (e.g., glutamate, GABA, acetylcholine, monoamines and neuropeptides) play critical roles in the slower modulation of fast synaptic transmission and electrical activity (*Elphick et al., 2018*; *Grimmelikhuijzen and Hauser, 2012*; *Jékely, 2013*; *Krishnan and Schiöth, 2015*; *Varoqueaux and Fasshauer, 2017*).

Because modulatory neuropeptides are not subject to the rapid transmitter re-uptake and/or degradation processes necessary for fast synaptic transmission, secreted neuropeptides are thought to persist long enough (e.g., minutes) in brain interstitial spaces for diffusion to very-high-affinity NP-GPCRs hundreds of micrometers distant from release sites (*Ludwig and Leng, 2006*; *Nässel, 2009*; *Russo, 2017*). Neuropeptide signaling can thus be presumed both 'paracrine', with secretion from individual neurons hitting receptor-positive cells over substantial diffusion distances and converging by diffusion from many local secreting neurons onto single receptor-positive neurons, and to be relatively slow (seconds-to-minutes timescale of action). Though present information is limited, eventual degradation by interstitial peptidases nonetheless probably restricts diffusion of most neuropeptides to sub-millimeter, local circuit distance scales.

The receptors encoded by NP-GPCR genes are highly diverse in ligand specificity but less diverse in downstream signaling impacts. Although GPCR signaling has long been recognized as complex and many faceted (*Hamm, 1998*), most known neuronal NP-GPCR actions reflect phosphorylation of ion channels, synaptic proteins or transcription factors mediated by protein kinases dependent on the second messengers cyclic AMP or calcium (*Mains and Eipper, 2006*; *Nadim and Bucher, 2014*; *van den Pol, 2012*). Primary effects of NP-GPCRs expressed in cortex, in turn, fall into just three major categories distinguished by G-protein alpha subunit (Gα) family. The Gi/o family (i/o) inhibits cAMP production, the Gs family (s) stimulates cAMP production, and the Gq/11 family (q/11) amplifies calcium signaling dynamics (*Syrovatkina et al., 2016*). For most NP-GPCR genes, the primary Gα family (e.g., i/o, s or q/11) is now known (*Alexander et al., 2017*) and offers a good first-order prediction of the encoded GPCR's signal transducing action. The profound functional consequences of neuromodulation by GPCRs range from modification of neuronal firing properties and calcium signaling dynamics through regulation of synaptic weights and synaptic plasticity (*Bargmann, 2012*; *Markram et al., 2013*; *McCormick and Nusbaum, 2014*).

It is well established that certain neuropeptides, including vasoactive intestinal peptide (VIP), somatostatin (SST), neuropeptide Y (NPY), substance P, and cholecystokinin (CCK), are detectable at high levels in particular subsets of GABAergic cortical neurons (*Tremblay et al., 2016*). These neuropeptides have come into broad use as markers for particular GABAergic interneuron classes, while the corresponding NPP and NP-GPCR genetics have provided molecular access to these and other broad neuron type classes (*Daigle et al., 2018*; *Maximiliano José et al., 2018*). In situ hybridization and microarray data, for example the Allen Brain Atlases (*Hawrylycz et al., 2012*; *Lein et al., 2007*), have also established that mRNA transcripts encoding these five NPPs and that many other NPPs and NP-GPCR genes are expressed differentially in many brain regions. There has been a critical lack, however, of comprehensive expression data combining whole-genome depth with single-cell

resolution. Absent such data, it has been difficult to generate specific and testable hypotheses regarding cortical neuropeptide function and to design robust experiments to test those hypotheses (*Tremblay et al., 2016*; *van den Pol, 2012*).

Here we describe new findings regarding NPP and NP-GPCR gene expression in mouse cortex. These findings have surfaced during a focused analysis of a previously published single-cell RNA-seq data acquired from large numbers of isolated mouse cortical neurons (*Tasic et al., 2018*). We begin by leveraging only the genomic depth and single-cell resolution of this dataset. Next, we introduce the transcriptomic neurotaxonomy developed in the same resource publication and explore the additional analytical power of a neurotaxonomic framework. Then, we distill these findings into specific and testable predictions concerning intracortical peptidergic modulation networks. Finally, we discuss the potential of a neurotaxonomically integrated view of neuromodulatory and synaptic networks to reveal previously obscure principles of cortical sensory, mnemonic and motor function.

## Results

The present study is based on analysis of a resource single-cell RNA-seq dataset acquired at the Allen Institute (*Tasic et al., 2018*) and available for download at http://celltypes.brain-map.org/rna-seq/. These RNA-seq data were acquired from a total of 22,439 isolated neurons, with detection of transcripts from a median of 9462 genes per cell  (min = 1445; max = 15,338) and an overall total of 21,931 protein-coding genes detected. Neurons were sampled from two distant and very different neocortical areas: 13,491 neurons from primary visual cortex (VISp), and 8948 neurons from anterior lateral motor cortex (ALM). Single neuron harvesting methods were designed to mildly enrich samples for GABAergic neurons, such that the sampled neuron population is roughly half GABAergic (47%) and half glutamatergic (53%). The resource publication (*Tasic et al., 2018*) should be consulted for full details of neuron harvesting, sample preparation, sequencing and data processing. Since we refer very frequently here to this resource publication and dataset, we'll refer to both now simply as 'Tasic 2018', and all further references here to neuron 'class', 'subclass' or 'type' should be understood to refer specifically to the particular mouse neocortex neurotaxonomy described in the Tasic 2018 publication.

The Tasic 2018 single-cell RNA-seq data tables report the abundance of transcripts from individual neurons in both 'counts per million reads' (CPM) and 'fragments per kilobase of exon per million reads mapped' (FPKM) units. Our analysis of this data compares gene expression levels quantitatively, with two distinct use cases: (1) comparisons across large sets of different genes, and (2) comparisons of the same gene across different individual cells, cell types and brain areas. We have relied upon FPKM data (*Mortazavi et al., 2008*; *Pimentel, 2014*), for use case 1 (i.e., the *Tables 1* comparisons across genes). For use case 2 (as in all figures below), we have preferred the CPM units, because these units were used to generate the Tasic 2018 neurotaxonomy. In any case, choice of CPM vs. FPKM units would have very little impact on the present outcomes.

### Single-neuron expression profiles of 18 select neuropeptide precursor (NPP) genes

*Table 1* lists results of analyzing the expression of 18 NPP genes in all 22,439 individual neurons represented in Tasic 2018. Here we have made use of the 'peak FPKM' (pFPKM) metric described in Materials and methods below to quantify the expression of specific genes in highly expressing subsets of single-cell populations that exhibit highly variant expression of that particular gene. Each of the 18 NPP genes on this list meets two conditions: (1) the included NPP gene is highly expressed (top quintile pFPKM over all protein-coding genes) across VISp and ALM cortical areas, and (2) at least one gene for an NP-GPCR selective for the predicted product of at least one of the 18 NPPs is highly expressed in neurons within the same local area of neocortex (see Table 2). The first requirement was imposed to increase the likelihood of active secretion of the NP product encoded by the candidate NPP gene. The second requirement, for 'cognate' pairing between each included NPP and a locally expressed NP-GPCR gene, was imposed to elevate the likelihood of paracrine NP signaling within a cortical local circuit volume, as envisioned in Introduction above. The process for selection of these 18 NPP genes is described in more detail in Materials and methods. *Table 1* lists Peak FPKM values for each of the 18 NPP genes, percentile and absolute ranks of that Peak FPKM value across all protein-coding genes, the percentage of cells sampled in which expression of the

**Table 1.** Single-cell RNA-seq expression statistics for 18 highly expressed neuropeptide precursor protein (NPP) genes cognate to locally expressed NP-GPCR genes (see **Table 2**).

NPP genes are tabulated here along with peak single-cell expression levels as pFPKM (Peak FPKM, see Materials and methods), percentile and absolute ranking of these pFPKM values across pFPKMs for all 21,931 protein-coding genes, and the percentage of cells sampled in which transcripts of the given NPP gene were detected at > 1 CPM. The table also lists predicted neuropeptide products, and genes encoding the locally expressed G-protein-coupled receptors (NP-GPCRs) cognate to the NPP (see **Table 2**). NPP genes are listed here in descending order of Peak FPKM. Pastel color fills in the 'Cognate NP-GPCR Genes' column correspond to i/o (pink), s (light green) and q/11 (light blue) transduction families of associated G-protein and will be used to highlight these families consistently in all following figures.

| NPP Gene | Peak FPKM | pFPKM Percentile | pFPKM Rank | % Cells | Predicted Neuropeptides | Cognate NP-GPCR Genes | |
|----|----|----|----|----|----|----|----|
| Npy | 108,865 | 100.00 | 1 | 42 | Neuropeptide Y | Npy1r, Npy2r, Npy5 | |
| Sst | 70,274 | 99.99 | 2 | 26 | Somatostatins | Sstr1, Sstr2, Sstr3, Sstr4 | |
| Vip | 48,747 | 99.99 | 3 | 33 | Vasoactive Intestinal Peptide | Vipr1, Vipr2 | |
| Tac2 | 18,284 | 99.98 | 4 | 15 | Neurokinin B | Tacr3 | |
| Cck | 16,396 | 99.97 | 6 | 69 | Cholecystokinins | Cckbr | |
| Penk | 11,160 | 99.96 | 8 | 26 | Enkephalins | Oprd1, Oprm1 | |
| Crh | 9,118 | 99.95 | 10 | 17 | Corticotropin-Releasing Hormone | Crhr1, Crhr2 | |
| Cort | 7,477 | 99.93 | 15 | 32 | Cortistatin | Sstr1, Sstr2, Sstr3, Sstr4 | |
| Tac1 | 5,728 | 99.92 | 18 | 11 | Substance P, Neurokinin A | Tacr1 | |
| Pdyn | 2,813 | 99.69 | 68 | 8 | Dynorphins | Oprd1, Oprk1, Oprm1 | |
| Pthlh | 1,656 | 99.29 | 156 | 18 | Parathyroid-Hormone-Like Hormone | Pth1r | |
| Pnoc | 698 | 97.68 | 509 | 23 | Nociceptins | Oprl1 | |
| Trh | 510 | 96.51 | 766 | 3 | Thyrotropin-Releasing Hormone | Trhr, Trhr2 | |
| Grp | 435 | 95.59 | 968 | 12 | Gastrin-Releasing Peptide | Grpr | |
| Rln1 | 258 | 91.99 | 1757 | 7 | Relaxin 1 | Rxfp1, Rxfp2 | Rxfp3 |
| Adcyap1 | 165 | 87.29 | 2788 | 26 | Adenylate Cyclase-Activating Polypeptides | Adcyap1r1, Vipr1, Vipr2 | |
| Nts | 121 | 82.14 | 3917 | 1 | Neurotensin | Ntsr1, Ntsr2 | |
| Nmb | 112 | 80.53 | 4270 | 14 | Neuromedin B | Nmbr | |

listed gene is detectible, predicted neuropeptide product(s) encoded, and the NP-GPCR gene(s) fulfilling requirement (2) for that NPP gene. Gene ontology results for the 18 select NPP genes are provided by **Supplementary file 1**. The Peak FPKM ranking columns in **Table 1** show that expression levels of most of the 18 NPP genes are extremely high in the range of values for all 21,931 protein-coding genes detected in all 22,439 neurons sampled. Of these genes, *Npy, Sst, Vip* and *Tac2* rank as the top four overall in pFPKM values, while *Cck, Penk* and *Crh* also rank in the top ten. Eleven of these NPP genes rank in the top percentile and all 18 rank in the top quintile by pFPKM. To the simplest first approximation, very high abundance of a given protein-coding transcript implies the potential, at least, for a very high rate of synthesis of the encoded protein. The extremely high peak abundance of these NPP transcripts thus suggests that NP precursor proteins could be synthesized at very high rates in neurons exhibiting such peak abundance. In a steady state, a high rate of synthesis would then necessarily imply a correspondingly high overall rate of protein product elimination. For an NP precursor protein, processing and secretion of active neuropeptide would seem the obvious and most likely route of elimination. The high abundance of transcripts encoding these 18 NPPs might thus be construed as *prima facie* evidence for robust secretion of neuropeptide products.

**Figure 1A** quantifies differential expression of the 18 NPP genes listed in **Table 1**. Each of the 18 color-coded solid curves represents a distribution of single-neuron CPM values for one NPP gene. Curves were generated by plotting CPM for each individual neuron in descending rank order along a cell population percentile axis. Each curve exhibits a transition from high to very low (commonly

**Table 2.** Single-cell RNA-seq expression statistics for 29 neuropeptide-selective, G-protein-coupled receptor (NP-GPCR) genes cognate to locally expressed NPP genes (see *Table 1*).

NP-GPCR gene peak FPPM values, percentile ranking, and percentage sampled as for NPP genes in *Table 1*. The table names encoded NP-GPCR proteins, A-F class of NP-GPCR, primary Gα signal transduction family (*Alexander et al., 2017*) and cognate NPP genes. Color fill in 'primary Gα family' column as in *Table 1*.

| NP-GPCR Gene | Peak FPKM | pFPKM Percentile | % Cells | Neuropeptide Receptor | GPCR Class | Primary Gα Family | Cognate NPP Genes |
|---|---|---|---|---|---|---|---|
| *Sstr2* | 413 | 95.3 | 42 | Somatostatin Receptor 2 | A4 | Gi/o | *Sst, Cort* |
| *Npy2r* | 291 | 93.1 | 10 | Neuropeptide Y Receptor Y2 | A9 | Gi/o | *Npy* |
| *Npy1r* | 272 | 92.4 | 50 | Neuropeptide Y Receptor Y1 | A9 | Gi/o | *Npy* |
| *Grpr* | 231 | 91 | 10 | GRP Receptor | A7 | Gq/11 | *Grp* |
| *Cckbr* | 210 | 90 | 52 | Cholecystokinin B Receptor | A6 | Gq/11 | *Cck* |
| *Ntsr2* | 161 | 86.9 | 17 | Neurotensin Receptor 2 | A7 | Gq/11 | *Nts* |
| *Npy5r* | 152 | 86.1 | 28 | Neurpeptide Y Receptor Y5 | A9 | Gi/o | *Npy* |
| *Nmbr* | 123 | 82.4 | 8 | Neuromedin B Receptor | A7 | Gq/11 | *Nmb* |
| *Rxfp1* | 121 | 82 | 22 | Relaxin Family Receptor 1 | A5 | Gs | *Rln1* |
| *Sstr4* | 106 | 79.5 | 28 | Somatostatin Receptor 4 | A4 | Gi/o | *Sst, Cort* |
| *Trhr* | 101 | 78.4 | 3 | TRH Receptor | A7 | Gq/11 | *Trh* |
| *Sstr1* | 90 | 76 | 38 | Somatostatin Receptor 1 | A4 | Gi/o | *Sst, Cort* |
| *Adcyap1r1* | 89 | 75.8 | 71 | ADCYAP1 Receptor 1 | B1 | Gs | *Adcyap1, Vip* |
| *Crhr1* | 86 | 74.9 | 28 | CRH Receptor 1 | B1 | Gs | *Crh* |
| *Rxfp3* | 85 | 74.7 | 5 | Relaxin Family Receptor 3 | A5 | Gi/o | *Rln1* |
| *Oprl1* | 82 | 73.8 | 48 | Opioid Receptor-Like 1 | A4 | Gi/o | *Pnoc* |
| *Crhr2* | 72 | 70.7 | 3 | CRH Receptor 2 | B1 | Gs | *Crh* |
| *Tacr3* | 65 | 68 | 3 | Tachykinin Receptor 3 | A9 | Gq/11 | *Tac2* |
| *Oprk1* | 64 | 67.4 | 3 | Kappa-Opioid Receptor | A4 | Gi/o | *Pdyn* |
| *Tacr1* | 56 | 64.2 | 3 | Tachykinin Receptor 1 | A9 | Gq/11 | *Tac1* |
| *Pth1r* | 51 | 61.6 | 15 | PTH 1 Receptor | B1 | Gq/11 | *Pthlh* |
| *Vipr1* | 41 | 56.1 | 28 | VIP Receptor 1 | B1 | Gs | *Vip, Adcyap1* |
| *Oprm1* | 35 | 52.1 | 43 | Mu-Opioid Receptor | A4 | Gi/o | *Penk, Pdyn* |
| *Trhr2* | 30 | 48.9 | 10 | TRH Receptor 2 | A7 | Gq/11 | *Trh* |
| *Vipr2* | 30 | 48.4 | 0.5 | VIP Receptor 2 | B1 | Gs | *Vip, Adcyap1* |
| *Rxfp2* | 28 | 47.3 | 4 | Relaxin Family Receptor 2 | A5 | Gs | *Rln1* |
| *Oprd1* | 26 | 45.8 | 13 | Delta-Opioid Receptor | A4 | Gi/o | *Penk, Pdyn* |
| *Ntsr1* | 24 | 44.3 | 10 | Neurotensin Receptor 1 | A7 | Gq/11 | *Nts* |
| *Sstr3* | 17 | 39.5 | 21 | Somatostatin Receptor 3 | A4 | Gi/o | *Sst, Cort* |

zero) expression across the sampled neuron population, but these transitions occur over very different population percentile ranges, providing clear evidence for highly differential single-cell expression of these genes. Percentages of the sampled neuron population with detectable expression of a given NPP gene range from more than 65% for *Cck* down to ~1% for *Nts*. (Note, however, that the cell population sampled has been enriched for GABAergic cell types as described in Tasic 2018).

The RNA-seq data suggest that all, or at least very nearly all, neocortical neurons express at least one NPP gene. The dashed curve in *Figure 1A*, labeled 'Max NPP Gene', was generated by plotting CPM values of the NPP gene with the highest CPM in each individual cell in descending order along a cell population percentile axis. This curve therefore shows that 97% of the sampled mouse cortical neurons express at least one NPP gene at >1 CPM and that 80% express at least one NPP gene at >1,000 CPM, a very high level. When one takes into account the pulsatile nature of transcription

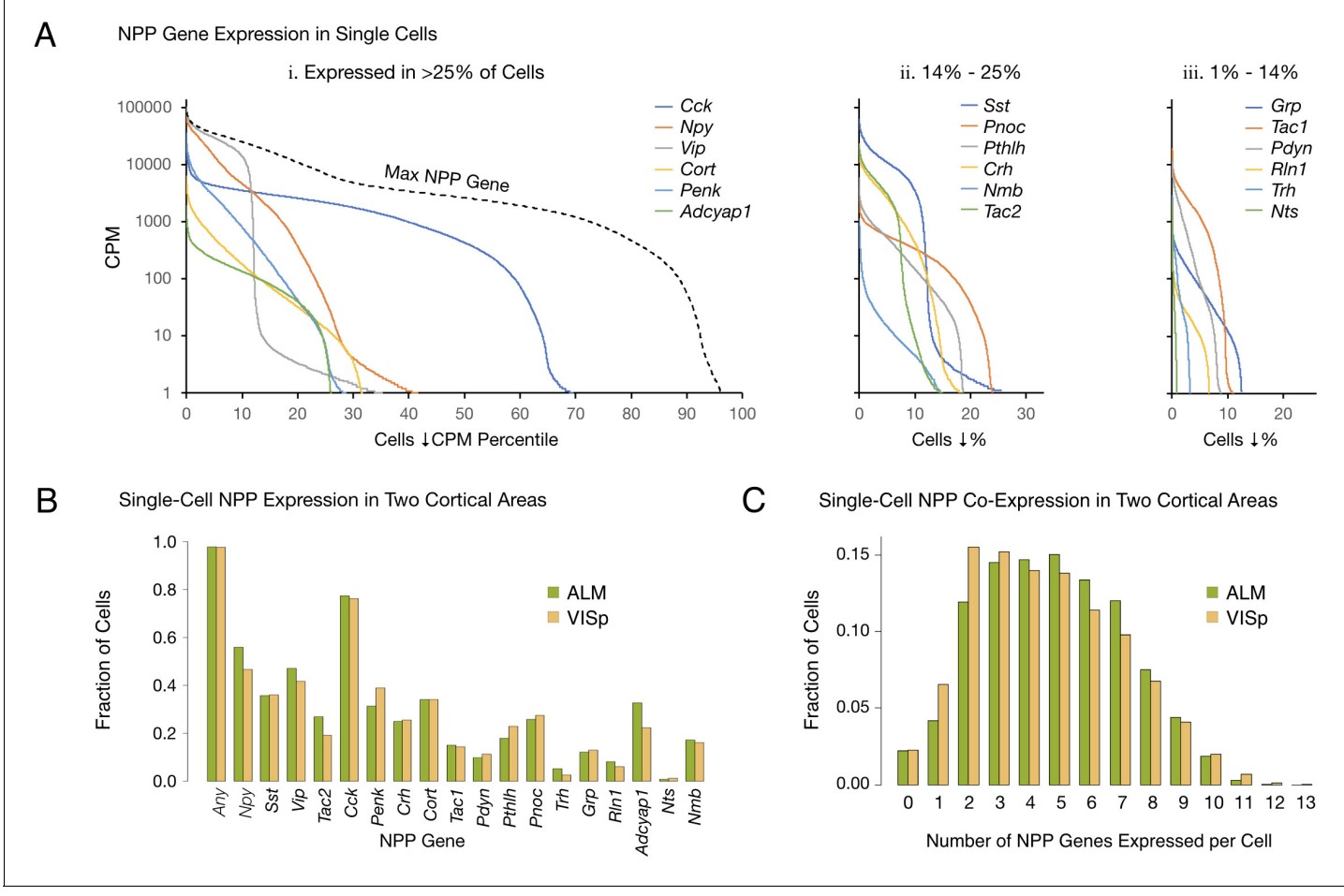

**Figure 1.** Single-cell NPP gene expression and co-expression statistics for distant neocortical areas VISp and ALM show that expression patterns for 18 NPP genes are highly differential within both neocortical areas but highly conserved between areas. (**A**) Different NPP genes show very different expression level distributions across the 22,439 VISp+ALM neurons sampled. Color-coded solid curves plot single-cell CPM values for the specified individual NPP genes in descending order along a cell population percentile axis. The 18 curves are segregated for clarity into three panels (I, ii, iii) sorted by cell population percentiles at which CPM values fall below 1. Large differences in fractions of cells expressing different NPP genes are evident. The dashed curve labeled 'Max NPP Gene' in panel A.i was generated by plotting CPM values of for the most abundant NPP transcript in each individual cell in descending order. (**B**) Fractions of cells expressing each NPP genes represented separately for 13,491 VISp neurons and 8,948 ALM neurons, showing conservation between areas of the patterning of NPP expression fractions detailed in panel A. (**C**) Histograms illustrating frequencies of various multiples of NPP genes co-expressed in individual neurons, represented separately for VISp and ALM neurons. The paired vertical bars show strong conservation of co-expression patterns between the two areas.

The online version of this article includes the following source data for figure 1:

**Source data 1.** Sorted NPP CPM distributions for all neurons.
**Source data 2.** Single-cell NPP CPM expression table.

(*Suter et al., 2011*) and the stochastic nature of RNA-seq transcript sampling (*Fu and Pachter, 2016*; *Kim et al., 2015*; *Tasic et al., 2016*), these numbers might be best understood as lower limits. The results summarized in *Figure 1A* may therefore be consistent with the proposition that every cortical neuron is peptidergic.

## Conservation of NPP gene expression statistics between VISp and ALM

The paired bars in *Figure 1B* represent fractions of cells expressing a given gene in each of the two cortical areas. It is obvious that the differential expression profiles in VISp and ALM are highly similar ($\rho$ = 0.972, p<1.72E-11), in spite of stark differences in function and cytoarchitecture between these two areas. Conservation of expression fractions across so many genes in such divergent cortical

areas suggests that these patterns have strong connections to conserved features of cortical function and argues against these patterns being secondary to more ephemeral variables such as neuronal activity patterns, which seem unlikely to be highly conserved between VISp and ALM areas. *Figure 1C* represents frequencies with which transcripts of various multiples drawn from the set of 18 NPP genes were detected in individual neurons. These data establish a remarkable degree of NPP gene co-expression in almost all individual cortical neurons. The modal number of co-expressed NPP genes detected is two in VISp and five in ALM, but both distributions are actually quite flat between 2 and 5, with broad plateaus out to seven co-expressed NPP genes per cell and a substantial tail out to 10. *Figure 1C* also reveals strong similarities of NPP co-expression distributions between VISp and ALM.

## Single-neuron expression profiles of 29 select neuropeptide receptor (NP-GPCR) genes

*Table 2* lists 29 NP-GPCR genes that are highly expressed in varied subsets of the 22,439 individual neurons sampled from cortical areas VISp and ALM. These 29 genes encode receptor proteins substantially selective for neuropeptide products encoded by the 18 NPP genes of *Table 1* (cross-referenced from that table as 'Cognate NP-GPCR Genes'). *Table 2* provides quantitative information on expression levels of these 29 NP-GPCR genes, names the receptor proteins they encode, indicates the A-F GPCR class and expected primary G$\alpha$ family and cross-references back to the cognate cortically-expressed NPP genes. As noted above, the 18 NPP genes and 29 NP-GPCR genes listed in *Tables 1* and *2* were selected for focused analysis here due to their cognate pairing and the consequent implication of local intracortical signaling. Methods of NP-GPCR gene selection are described more fully in Materials and methods. A more complete listing of NP-GPCR genes with pFPKM values in provided by Table 2— source data 1. Gene ontology results for the 29 select NPP genes are provided by *Supplementary file 2*. The 'pFPKM Percentile' column in *Table 2* shows that most of these 29 NP-GPCR genes are expressed in cortex with Peak FPKM values well above median (50th percentile) for all protein coding genes. The range of cortical neuron pFPKM values for NP-GPCR genes does not match the extreme heights noted for NPP genes, but this is as expected given that NP-GPCR gene products are thought to be durable cellular components, unlikely to be rapidly disposed by secretion as expected for NPP gene products. Peak FPKM values for NP-GPCR transcripts are nonetheless quite high in the range of transcripts of other durable cellular component genes, suggesting a strong likelihood that they are indeed translated into functionally important protein products.

The single-cell RNA-seq data expose very highly differential expression of NP-GPCR genes in cortical neurons. *Figure 2* represents expression patterns of the 29 NP-GPCR genes listed in *Table 2* in the same manner as for the 18 NPP genes in *Figure 1*. *Figure 2A* establishes that each of the 29 NP-GPCR genes is expressed in highly differential fashion across the 22,439 mouse cortical neurons sampled, as was the case for the 18 NPP genes. As was noted for NPP genes in *Figure 1*, each of the curves in *Figure 2A* exhibits a transition from very high to very low (commonly zero) expression across the sampled neuron population. These transitions occur at very different population percentile points, again providing clear evidence for highly differential expression. Percentages of the sampled neuron population expressing a given NP-GPCR gene (at greater than 1 CPM) range from more than 72% for *Adcyap1r1* down to 0.7% for *Vipr2*.

The RNA-seq data suggest that all, or at least very nearly all, neocortical neurons express at least one NP-GPCR gene. The dashed curve in the left panel of *Figure 2A*, generated similarly to the dashed curve for NPP genes in *Figure 1A*, shows that 98% of the sampled mouse cortical neurons express at least one NP-GPCR gene at >1 CPM and that 78% express at least one NP-GPCR gene at >100 CPM, lower than the comparable NPP curve in *Figure 1*, but still indicative of quite high expression. Again, these numbers must be understood as lower limits to percentages of cortical neurons actually expressing at least one of the 29 NP-GPCR genes, after taking into account the pulsatile transcription and stochastic sampling issues cited above. The results summarized in *Figure 2A* may thus be consistent with a conclusion that every cortical neuron expresses at least one NP-GPCR gene cognate to a cortically expressed NPP gene.

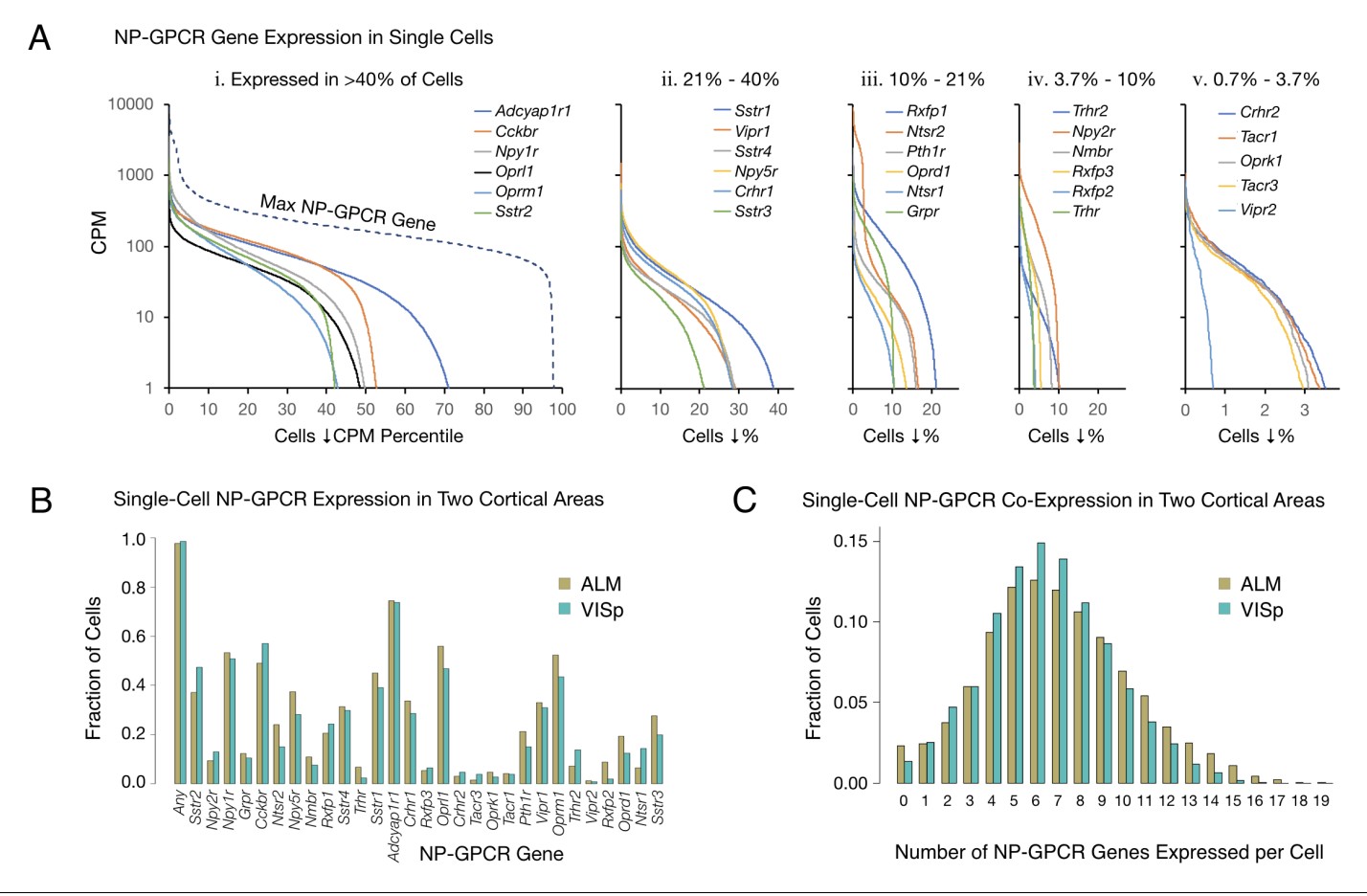

**Figure 2.** Single-cell NP-GPCR gene expression and co-expression statistics for distant neocortical areas VISp and ALM show that expression patterns for 29 NP-GPCR genes are highly differential within neocortical areas but conserved between areas. (A) Different NP-GPCR genes show very different expression level distributions across the 22,439 VISp+ALM neurons sampled. Color-coded solid curves plot single-cell CPM values for the specified individual NP-GPCR genes in descending order along a cell population percentile axis. The 29 curves are segregated for clarity into five panels (i-v) sorted by cell population percentiles at which CPM values fall below 1. Large differences in fractions of cells expressing different NP-GPCR genes are evident. Dashed curve labeled 'max NP-GPCR Gene' in panel A. i was generated by plotting CPM values of the highest CPM NP-GPCR gene for each individual cell in descending order. (B) Fractions of cells expressing each NP-GPCR genes represented separately for 13,491 VISp neurons and 8,948 ALM neurons, showing strong conservation between areas of the patterning of NP-GPCR expression fractions documented in panel A. (C) Histograms illustrating frequencies of various multiples of NP-GPCR gene co-expression in individual neurons, represented separately for VISp and ALM neurons. The paired vertical bars illustrate strong conservation of co-expression patterns between the two cortical areas.

The online version of this article includes the following source data and figure supplement(s) for figure 2:

**Source data 1.** Sorted NP-GPCR CPM distributions for all neurons.
**Source data 2.** Single-cell NP-GPCR CPM expression table.
**Figure supplement 1.** Co-expression of cognate NPP/NP-GPCR pairs.

## Conservation of NP-GPCR gene expression statistics between VISp and ALM

*Figure 2B* provides evidence for strong conservation of differential NP-GPCR expression profiles between distant cortical areas VISp and ALM. The paired bars represent fractions of cells expressing a given gene in each of the two areas, again revealing strong similarities of differential expression profiles in the two very different neocortical areas (ρ = 0.959, p<2.2E-16). *Figure 2C* represents frequencies of NP-GPCR gene co-expression multiples detected in individual neurons. These data establish that multiple NP-GPCR genes are co-expressed in almost all cortical neurons and that numbers of genes co-expressed are even higher than those noted above for co-expression of NPP genes. Modal numbers of co-expressed NP-GPCR genes detected is six in both VISp and ALM with

broad plateaus extending out to 12 co-expressed NP-GPCR genes per cell. The striking similarities of NP-GPCR co-expression distributions between the two otherwise divergent neocortical areas once again suggests that the patterning of NP-GPCR co-expression may have consequences for cortical function that are conserved because they are functionally important. As illustrated by *Figure 2—figure supplement 1*, it is furthermore common for individual neurons to co-express cognate NPP/NP-GPCR pairs, raising the intriguing possibility of cell-autonomous feedback mediated by an autocrine action of a secreted NP product on the secreting cell itself. *Figure 2—figure supplement 1* additionally shows that cognate pair co-expression patterning is also highly conserved between areas VISp and ALM.

## Neurotaxonomic profiling of NPP and NP-GPCR gene expression

The analysis so far has relied solely upon the genomic depth and single-cell resolution characteristics of the Tasic 2018 transcriptomic data. We now proceed to explore the analytical power of the transcriptomic neurotaxonomy developed as part of the Tasic 2018 study. This neurotaxonomy makes it possible to predict a protein 'parts list' for any neuron that can be mapped to a given transcriptomic taxon. Combined with tools for genetic access to transcriptomic taxa, transcriptomic taxonomy thereby offers rich prospects for experimental test of such predictions (see also Discussion below), The present analysis will make extensive use of the Tasic 2018 neurotaxonomy's representation of 115 glutamatergic and GABAergic transcriptomic neuron types (see *Figure 3—figure supplement 1*).

Figure 3 shows transcriptomic gene expression 'heatmaps', representing transcript abundance for each of 18 NPP (*Figure 3A*) and 29 NP-GPCR (*Figure 3B*) across all 115 glutamatergic and GABAergic neuron types by log10-scaled pseudocolor. These heatmaps show that expression of every one of these 47 genes is highly specific to particular neuron types, but that type specificity varies greatly from gene to gene. Note that CPM expression levels vary across neuron types by factors exceeding 10,000 for many NPP genes and 1000 for many NP-GPCR genes. These heat maps also show that every neuron type expresses multiple NPP and NP-GPCR genes and that each of the NPP and NP-GPCR genes is expressed in multiple neuron types (with Vipr2 in one Pvalb type as a near exception). These two heatmaps further show many cases where both an NPP gene and its cognate NP-GPCR receptor are expressed in the same neuron type, with the *Cck/Cckbr* and *Adcyap1/Adcyap1* r1 pairs being particularly prominent examples. Quite intriguingly, these expression heat maps also suggest that each of the neuron types might be distinguished by a unique pattern of expression of these 47 NP genes. This possibility will be explored quantitatively in connection with *Figure 4* below.

The dashed vertical line spanning *Figure 3A and B* heatmaps divides glutamatergic and GABAergic neuron types and provides for ready comparison of NP gene expression patterns in these two broad neurotaxonomic classes. *Figure 3A* shows clearly that more NPP genes are expressed more strongly in GABAergic than in glutamatergic types. This differential is consistent with a long history of neuroscientific use of neuropeptide products as protein markers of GABAergic neuron subsets (e. g., VIP, SST, NPY, Substance P), which has no parallel in the marking of glutamatergic neuron subsets. *Figure 3A* nonetheless also shows that every glutamatergic type expresses at least one NPP genes at a very substantial level. *Figure 3B* shows that the broader expression of NPP genes in GABAergic over glutamatergic types is leveled or even reversed for NP-GPCR genes. That is, while GABAergic neurons clearly show the more prolific and varied expression of NPP genes, glutamatergic neurons may be somewhat more prolific expressors of NP-GPCR genes.

Additional graphics on the *Figure 3* heatmaps further delineate the Tasic 2018 neurotaxonomy. A cladogram reflects the hierarchical similarity progression from the broad GABAergic and glutamatergic classes to the 115 individual neuron types, as aggregated across VISp and ALM cortical areas. Tinted rectangles and labels call out the five glutamatergic and seven GABAergic subclasses (see also *Figure 3—figure supplement 1*). Thin gray vertical lines crossing both NPP and NP-GPCR heatmaps demarcate those same subclasses. This delineation of subclasses shows that expression of some genes tends to remain constant within some subclasses, but to change abruptly at subclass boundaries. This does not seem, however, to be a very general case. Many genes show expression that varies widely by type within subclass. *Figure 3C* quantifies such residual expression variation for all NPP and NP-GPCR genes within each subclass. These significant residuals justify the use of more narrowly defined taxa (e.g., the 115 neuron types) to adequately characterize cortical neuropeptide

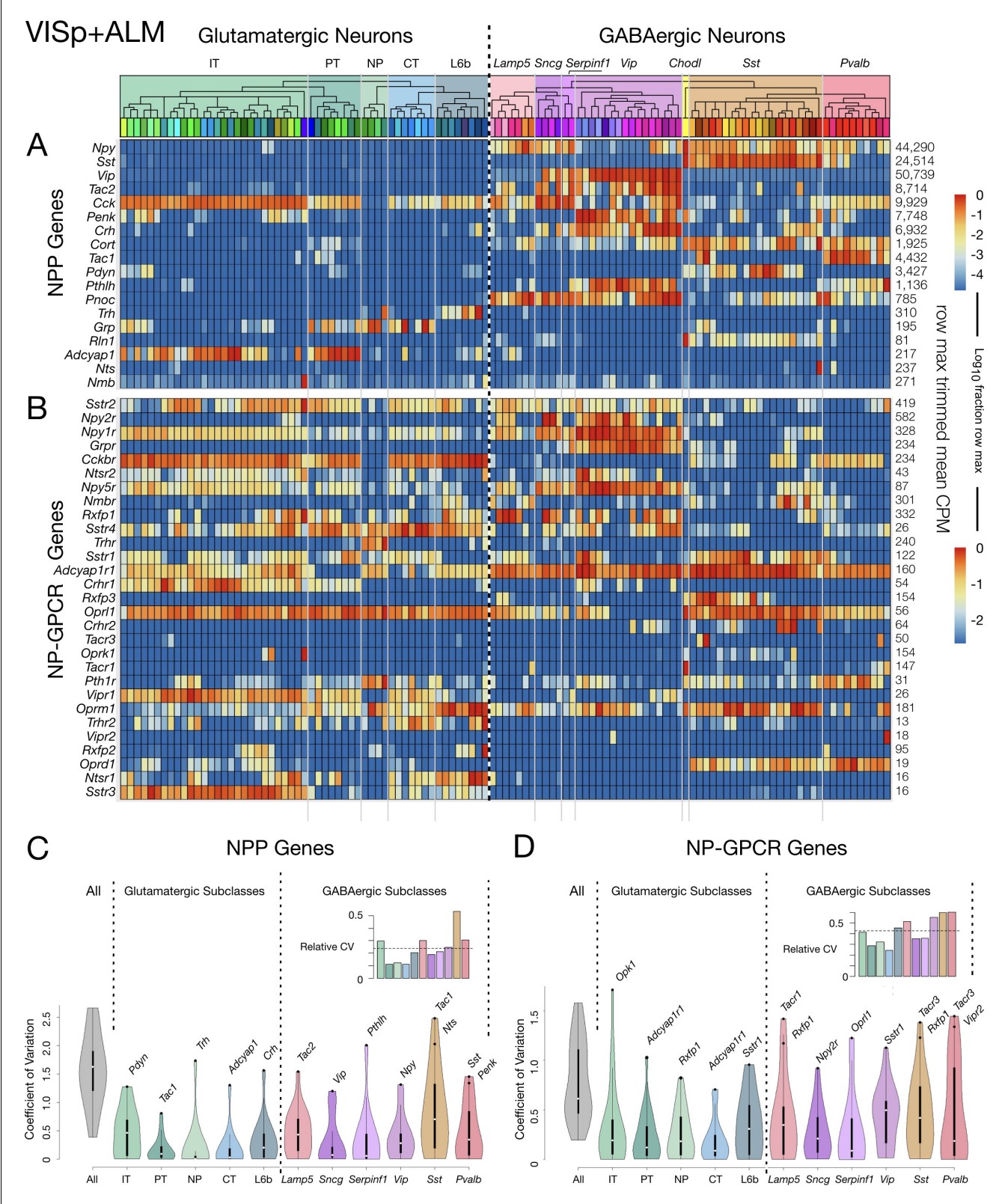

**Figure 3.** Neurotaxonomic heatmaps reveal highly neuron-type-specific expression of (**A**) 18 NPP and (**B**) 29 NP-GPCR genes in 22,439 individual neurons harvested from areas VISp and ALM. Trimmed-mean (5% trim) CPM expression values for each of the 115 VISp+ALM glutamatergic and GABAergic neuron types (see *Figure 3—figure supplement 1*) are normalized per gene to maximum value indicated at right for each row and pseudocolored according to log10 scales at right. Note that these scales represent 5 (NPP) and 3 (NP-GPCR) orders of magnitude and that each gene
*Figure 3 continued on next page*

*Figure 3 continued*

spans the entire pseudocolor range across neuron types. Subclasses are called out here by labels (IT, PT, NP, CT, L6b for glutamatergic types; Lamp5, Sncg, Serpinf1, Vip, Chodl, Sst, Pvalb for GABAergic types) and demarcated on the heatmaps by thin gray lines. Gene rows are ordered here as in *Tables 1* and *2*. (C) Violin plots representing coefficients of CPM variation (CV) for 18 NPP genes across types pooled within each of the 11 subclasses indicated (Chodl not represented here as it is a singular neuron type) and globally across all cell types ('All'). Callouts on each violin indicate genes of highest CVs within each subclass. Inset shows within-subclass CV/global CV demonstrating variation within subclasses is a significant fraction of global variability (dotted line mean = 0.239). See *Figure 3—figure supplement 2* for individual gene statistics. (D) Similar for 29 NP-GPCR genes showing greater relative variability, mean CV = 0.427. See *Figure 3—figure supplement 3* for individual gene statistics.

The online version of this article includes the following source data and figure supplement(s) for figure 3:

**Source data 1.** Cell types, cluster ids and color codes for ALM and VISp regions.
**Figure supplement 1.** Tasic 2018 neurotaxonomy.
**Figure supplement 2.** NPP expression variation within subclasses.
**Figure supplement 3.** NP-GPCR expression variation within subclasses.

gene expression. Relationships between NP gene expression patterns and the Tasic 2018 neurotaxonomy will be examined more quantitatively in the following section.

## Transcripts of 18 NPP and 29 NP-GPCR genes are exceptionally potent neuron-type markers

The strong marker patterning of the 47 NP gene expression profiles evident in *Figure 3* suggests the possibility that each of the 115 glutamatergic and GABAergic neuron types might be distinguished by a unique combination of these 18 NPP and 29 NP-GPCR genes. To explore this possibility and compare NP transcriptomes to other transcriptome subsets quantitatively, we developed the analysis presented in *Figure 4*.

We began by asking whether there exists a low dimensional representation of gene expression that naturally separates neurons of different types into distinct parts of that low-dimensional space. The extent to which a neuron's location in such a space can be inferred from the expression of a limited subset of genes (such as the 47 NP genes) would then provide a measure of sufficiency of that subset to determine the type of that neuron accurately. Hierarchical clustering methods to define neuron types based upon gene expression are well established (*Hastie et al., 2001*; *Oyelade et al., 2016*) but have difficulty when comparing and making inferences between datasets. We therefore devised a machine learning approach based on linking deep neural networks called autoencoders (*Hinton and Salakhutdinov, 2006*) to address this question explicitly and quantitatively.

We trained a single autoencoder network to represent cells in a low dimensional space based on CPM values of the 6083 most highly expressed genes (HE genes) in the Tasic 2018 dataset. *Figure 4A* shows the result of one such two dimensional encoding, where each of the 22,439 individual neurons appear as a distinct dot colored by its type assignment. The tight grouping of type-coding colors evident in *Figure 4A* implicitly conveys that position within this latent space corresponds well to neuron types, despite the fact that the autoencoder did not have prior information about the Tasic 2018 classification. With the first autoencoder held as fixed, we trained a second autoencoder, linked to the first, to obtain a low-dimensional representation based on a much smaller subset of genes. *Figure 4B* shows a two-dimensional representation of the same 22,439 neurons based on 47 NP genes. Again the tight color grouping suggests that the 47 NP genes alone suffice to assign types in close register to the Tasic 2018 neurotaxonomy. The autoencoder network architectures are schematized in *Figure 4C*. The cost function used to train the second autoencoder included a penalty term to minimize differences in the representation of cells compared to that obtained by the first autoencoder. This was done to ensure that the latent spaces of the two autoencoders are as similar as possible while faithfully representing the expression patterns of the respective gene sets they receive as input. This procedure allowed us to visualize the similarity between the gene sets in a latent space that captures type information, and to quantify the extent to which any small gene subset by itself could be used to identify neuron types.

To quantify the type classification ability of different gene sets, we used Quadratic Discriminant Analysis (QDA) (*Hastie et al., 2001*) to perform supervised classification using five-dimensional latent space representations of the different gene sets obtained by autoencoder networks. We obtained a measure on a per-cell basis, resolution index (RI) to evaluate the degree of

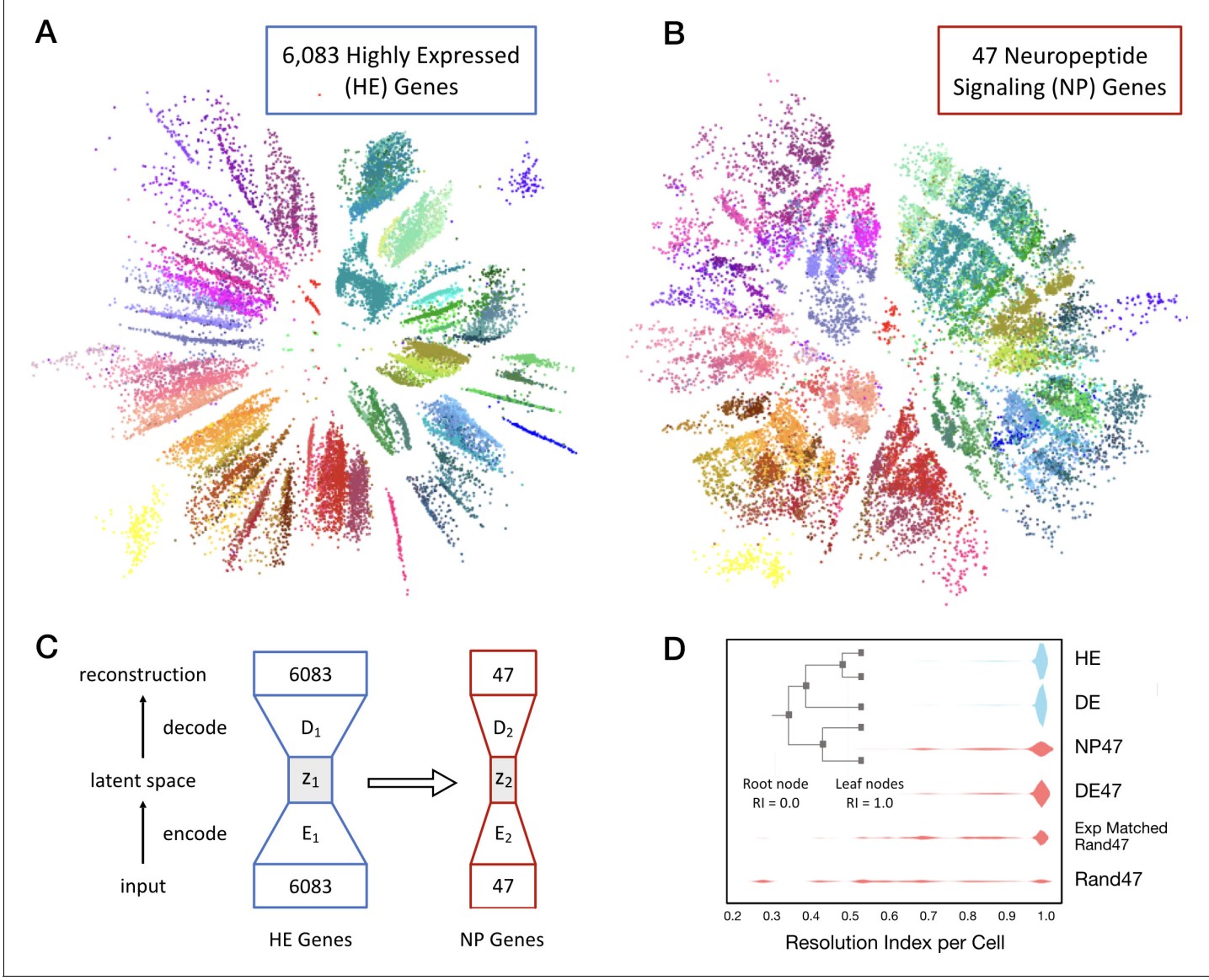

**Figure 4.** Neurons can be clustered effectively based on just 47 NP genes (18 NPP and 29 NP-GPCR). (**A**) A two-dimensional latent space representation of 22,439 cells based on 6083 highly expressed (HE) genes obtained by an autoencoding neural network. Dots represent individual cells, colored according to the type-code assignments of *Tasic et al. (2018)* (see *Figure 3—figure supplement 1*). Cells of the same type appear as grouped into distinct islands, which suggests that classifiers trained to identify cell types would perform well with such low dimensional representations of gene expression. (**B**) Two-dimensional representation of neurons in such a latent space $z_2$, based on the 47 NP genes. (**C**) Schematic of the network architecture used to train the second autoencoder that learns to represent neurons in a latent space z2 that is similar to z1. This second autoencoder represents cells in the latent space based on much smaller gene sets. (**D**) Inset illustrates resolution index (RI) associated with nodes on the hierarchical tree used in the per-cell RI calculation. RI distribution (see Materials and methods) for NP genes-based cell type classification shows that a vast majority of the cells can be correctly classified up to the type level (leaf nodes, RI = 1.0) of the Tasic 2018 hierarchy. Errors in classification (RI <1.0) at the type level are nevertheless resolved at the class level of the hierarchy, as indicated by the high values for RI for the remaining cells. High average RI for HE genes, and 4020 differentially expressed (DE) genes, and 47 DE genes indicates that the cell type classification procedure based on autoencoder representations is accurate. The average RI for cell type classification based on the 47 NP genes is significantly higher (p<0.01, bootstrap) than both, subsets of 47 genes selected randomly (Rand47, n = 100 subsets), and selected randomly but with expression levels matched to the NP genes (Rand47 ExpMatch, n = 100 subsets).

The online version of this article includes the following figure supplement(s) for figure 4:

**Figure supplement 1.** Autoencoder expression clustering performance by neuron type.

**Figure supplement 2.** 1'''Determination of optimal autoencoder latent space dimensionality.

correspondence of classification results with the Tasic 2018 neurotaxonomy. The resolution index averaged over all cells is used as a summary statistic to quantify the ability of different gene sets in resolving neuron types. Briefly, QDA was performed iteratively on a given latent space representation, starting with all the leaf node type labels of the neurotaxonomy. In each subsequent iteration the number of labels was reduced by successively merging leaf node labels into their parent node class label (inset, *Figure 4D*). RI = 1.0 for a neuron that is assigned the correct type (e.g., Pvalb Reln Tac1) and 0.0 < RI < 1.0 for neurons for which the iterative QDA based classification could determine the correct label only up to a subclass (e.g. Pvalb). A neuron is assigned RI = 0.0 if the QDA-based classification failed to determine the correct label even at the glutamatergic or GABAergic level.

*Figure 4D* shows neuron type classification results based on five dimensional latent space representations of different subsets of genes (k = 13 fold cross validation). For the 6,083 HE genes and a set of 4020 genes most differentially expressed (DE genes) across neuron types, the latent space is obtained with the first autoencoder, and the RI distributions shown in blue have average values of 0.986 and 0.987, respectively, close to the theoretical maximum of 1.0 that can only be achieved with perfect type classification for all neurons in the dataset. For subsets of 47 genes, the latent representations were obtained with the second linked autoencoder, and the corresponding RI distributions are colored red. A set of 47 DE genes achieves average RI = 0.964. These results confirm the idea that autoencoder-based low dimensional representations of gene expression can be used for accurate type classification. The 47 NP genes can be used to classify neuron types well, with average RI = 0.925 and a majority of the neurons (62%) classified correctly at the type level (with nearly uniform performance across all neuron types, see *Figure 4—figure supplement 1*). This RI performance is significantly higher (p<0.01, bootstrap) than the average RI for of subsets of genes chosen randomly (0.641 ± 0.047, n = 100), and chosen randomly but with expression levels matched with the NP genes (0.843 ± 0.027, n = 100), with none of the individual randomly selected subsets reaching the NP gene index of 0.925. Note that genes in the 47 DE set were chosen with prior knowledge of the Tasic 2018 taxonomy, while the 47 NP gene set was not. This distinction thus makes the near match of the 47 NP to the 47 DE gene sets in average RI all the more striking. This demonstration of the exceptional power of NP genes to mark transcriptomic neuron types reinforces earlier indications of an especially close and fundamental connection between neuropeptide gene expression and neuron type identity.

## Conservation of NPP and NP-GPCR gene expression profiles between VISp and ALM

*Figure 5* juxtaposes separate VISp and ALM expression profiles for NPP and NP-GPCR genes across 93 VISp neuron types (*Figure 5A*) and 84 ALM neuron types (*Figure 5B*). Similarities of expression profiles for the two areas are obvious in *Figure 5*, but there are also visible differences. The latter are rooted primarily in the substantial divergence of glutamatergic neuron taxonomies discussed at length in *Tasic et al. (2018)*. Very strong similarities of both NPP and NP-GPCR expression profiles are most obvious for the GABAergic types, where the taxonomies are identical except for the absence of two GABAergic types in ALM (indicated by dark gray vertical placeholder bars in *Figure 5B*). The general conservation of neuron-type-specific expression patterns among common cell types between the two distant neocortical areas (NPP correlation: ρ = 0.974, p<2.2e-16, NP-GPCR: 0.877, p<2.2e-16) thus provides another indication of robust connection between NP gene expression and cortical neuron differentiation.

## Prediction of local peptidergic signaling from expression of cognate NPP/NP-GPCR pairs

Expression of an NPP gene in one neuron and a cognate NP-GPCR gene in another neuron nearby implies a possibility of directed paracrine signaling, with diffusion of a secreted peptide coupling the first neuron to the second. The present set of 47 cortical NP genes (18 NPP and 29 NP-GPCR) comprises the 37 distinct cognate NPP/NP-GPCR pairs enumerated in *Table 3* and predicts accordingly 37 distinct peptidergic neuromodulation networks. As noted in the Introduction, expected neuropeptide diffusion distances suggest that any neuron within a local cortical area (e.g., VISp or ALM) might signal by diffusion to any other neuron within that same local area, but almost surely not to

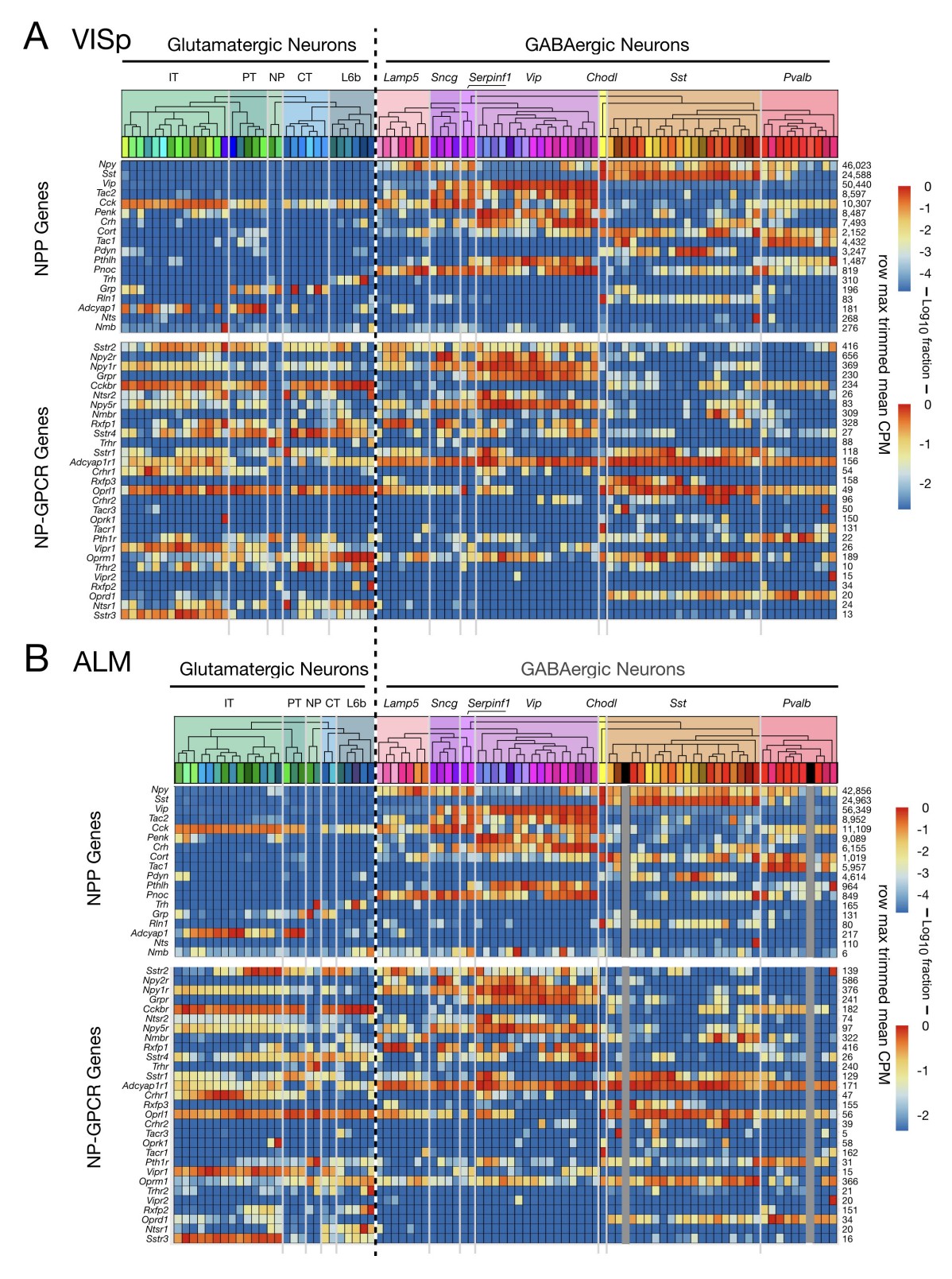

**Figure 5.** Neurotaxonomic heatmaps show strong conservation of NPP and NP-GPCR expression patterns between two distant neocortical areas. (**A**) Expression heatmap for 18 NPP and 29 NP-GPCR genes in 13,491 single VISp neurons classified by type. (**B**) Similar heatmap for 8948 single neurons harvested from ALM. Heatmaps generated and displayed as described in **Figure 3**, except for segregation here of VISp and ALM harvest areas. Heat maps are aligned horizontally here to match GABAergic neuron types between VISp and ALM. Vertical dark gray bars in **Figure 5B** are spacers marking

*Figure 5 continued on next page*

*Figure 5 continued*

the two GABAergic cell types absent in ALM. Glutamatergic neurotaxonomies are seen to differ substantially, but differences appear mainly at the finest, 'leaf' levels of the neurotaxonomic hierarchy (see Tasic 2018) and *Figure 3—figure supplement 1*).

The online version of this article includes the following source data for figure 5:

**Source data 1.** Cell types, cluster ids and color codes for VISp region.
**Source data 2.** Cell types, cluster ids and color codes for ALM region.

more distant areas (e.g., from VISp to ALM). In the following, we therefore make predictions of 74 (37 × 2) peptidergic distinct signaling networks, keeping separate consideration of signaling within VISp and within ALM.

## Prediction of peptidergic networks from neurotaxonomic NP gene expression profiles

*Figure 6* displays weighted adjacency matrix plots representing predictions of neuron-type-specific and neuron-subclass-specific peptidergic coupling from selections drawn from VISp and ALM of the 37 cognate NP gene pairs. The prediction matrices A-E are outer products (CPM*CPM units) of vectors representing expression (CPM units) of an NPP gene (columns) and a cognate NP-GPCR gene (rows) across all VISp or ALM neuron types. The predicted coupling matrices in F matrices are similar except that factor vectors are down-sampled by averaging neuron-type-specific CPM values within each of the subclasses (see Materials and methods for more details).

**Table 3.** The 18 NPP and 29 NP-GPCR genes of *Tables 1* and *2* constitute 37 cognate NPP/NP-GPCR pairs and predict at least 37 potentially distinct peptidergic modulatory networks.

The 37 pairs are enumerated here along with indications of the expected primary GPCR signal transduction class for each NP-GPCR (*Alexander et al., 2017*) and a fraction denoting frequency with which the given cognate pair occurs as a fraction of all neuron pairs surveyed. Pastel table fill colors denote G-protein transduction class as in *Tables 1* and *2*.

| # | Cognate Pair Symbol | NPP Gene | NP-GPCR Gene | Primary Gα Family | Fraction of Type Pairs | # | Cognate Pair Symbol | NPP Gene | NP-GPCR Gene | Primary Gα Family | Fraction of Type Pairs |
|---|---|---|---|---|---|---|---|---|---|---|---|
| 1 | Npy→Npy1r | Npy | Npy1r | Gi/o | 0.7805 | 19 | Vip→Vipr1 | Vip | Vipr1 | Gs | 0.496 |
| 2 | Npy→Npy2r | Npy | Npy2r | Gi/o | 0.341 | 20 | Vip→Vipr2 | Vip | Vipr2 | Gs | 0.052 |
| 3 | Npy→Npy5r | Npy | Npy5r | Gi/o | 0.8095 | 21 | Crh→Crhr1 | Crh | Crhr1 | Gs | 0.3925 |
| 4 | Sst→Sstr1 | Sst | Sstr1 | Gi/o | 0.751 | 22 | Crh→Crhr2 | Crh | Crhr2 | Gs | 0.2035 |
| 5 | Sst→Sstr2 | Sst | Sstr2 | Gi/o | 0.836 | 23 | Rln1→Rxfp1 | Rln1 | Rxfp1 | Gs | 0.2465 |
| 6 | Sst→Sstr3 | Sst | Sstr3 | Gi/o | 0.405 | 24 | Rln1→Rxfp2 | Rln1 | Rxfp2 | Gs | 0.07 |
| 7 | Sst→Sstr4 | Sst | Sstr4 | Gi/o | 0.806 | 25 | Adcyap1→Adcyap1r1 | Adcyap1 | Adcyap1r1 | Gs | 0.284 |
| 8 | Penk→Oprd1 | Penk | Oprd1 | Gi/o | 0.4955 | 26 | Adcyap1→Vipr1 | Adcyap1 | Vipr1 | Gs | 0.1465 |
| 9 | Penk→Oprm1 | Penk | Oprm1 | Gi/o | 0.9 | 27 | Adcyap1→Vipr2 | Adcyap1 | Vipr2 | Gs | 0.0155 |
| 10 | Cort→Sstr1 | Cort | Sstr1 | Gi/o | 0.6265 | 28 | Tac2→Tacr3 | Tac2 | Tacr3 | Gq/11 | 0.0955 |
| 11 | Cort→Sstr2 | Cort | Sstr2 | Gi/o | 0.6965 | 29 | Cck→Cckbr | Cck | Cckbr | Gq/11 | 0.6635 |
| 12 | Cort→Sstr3 | Cort | Sstr3 | Gi/o | 0.338 | 30 | Tac1→Tacr1 | Tac1 | Tacr1 | Gq/11 | 0.119 |
| 13 | Cort→Sstr4 | Cort | Sstr4 | Gi/o | 0.672 | 31 | Pthlh→Pth1r | Pthlh | Pth1r | Gq/11 | 0.392 |
| 14 | Pdyn→Oprd1 | Pdyn | Oprd1 | Gi/o | 0.2115 | 32 | Trh→Trhr | Trh | Trhr | Gq/11 | 0.016 |
| 15 | Pdyn→Oprk1 | Pdyn | Oprk1 | Gi/o | 0.0745 | 33 | Trh→Trhr2 | Trh | Trhr2 | Gq/11 | 0.055 |
| 16 | Pdyn→Oprm1 | Pdyn | Oprm1 | Gi/o | 0.4 | 34 | Grp→Grpr | Grp | Grpr | Gq/11 | 0.113 |
| 17 | Pnoc→Oprl1 | Pnoc | Oprl1 | Gi/o | 0.654 | 35 | Nts→Ntsr1 | Nts | Ntsr1 | Gq/11 | 0.0225 |
| 18 | Rln1→Rxfp3 | Rln1 | Rxfp3 | Gi/o | 0.106 | 36 | Nts→Ntsr2 | Nts | Ntsr2 | Gq/11 | 0.054 |
| | | | | | | 37 | Nmb→Nmbr | Nmb | Nmbr | Gq/11 | 0.5655 |

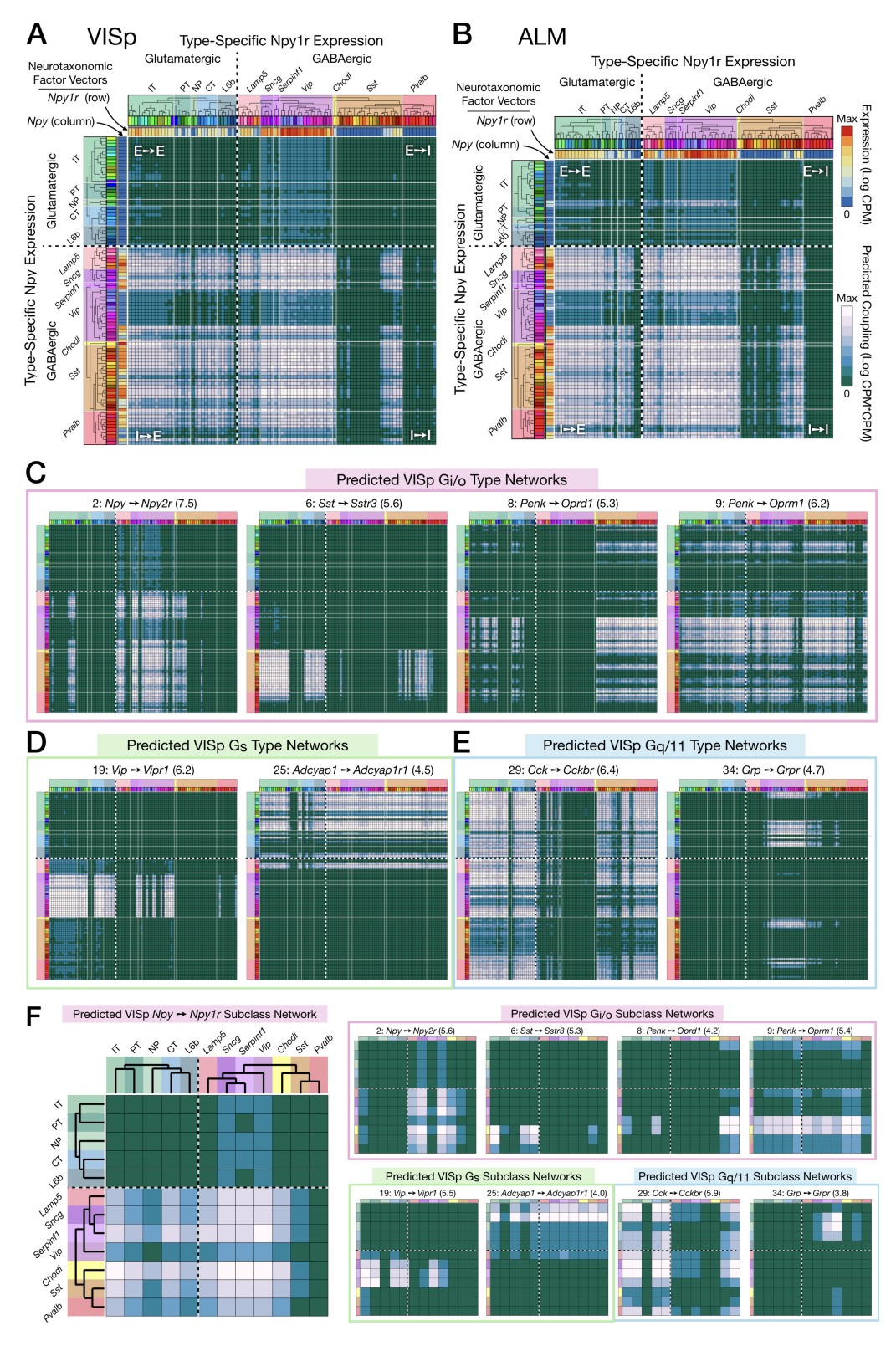

**Figure 6.** Neurotaxonomic expression profiles of 37 cognate NPP/NP-GPCR pairs predict 37 peptidergic networks. Weighted adjacency matrix plots predicting local peptidergic coupling amongst neuron types (A–E) and subclasses (F). Matrices were computed as outer products (CPM*CPM units) of row and column factor vectors representing abundance (CPM units) of NPP and cognate NP-GPCR genes. Pseudocolor scales representing both expression (CPM) and coupling (CPM*CPM) are logarithmic. (A) 93 × 93 matrix predicting coupling amongst 93 VISp neuron types based on type-

*Figure 6 continued on next page*

*Figure 6 continued*

specific expression of the Npy gene and the cognate Npy1r NP-GPCR gene, as indicated by row and column vector 'heat' strips called out by curved arrows. (B) An 84 × 84 square matrix similarly representing Npy-Npy1r coupling, depicted as in (A), except based only on the 84 ALM neuron types. Dashed crosses demarcate the four quadrants of directed NPP/NP-GPCR pairing between glutamatergic ('E') and GABAergic ('I') neuron types, called out as (E→ E), (E→ I), (I→ I) and (I→ E). Light gray lines, pastel color blocks and labels flanking both axes demarcate higher, subclass levels of the Tasic 2018 neurotaxonomy (as in *Figure 3A* above). (C–E) Exemplar matrix predictions for a further sampling of the 37 VISp cognate NPP/NP-GPCR pairs from each of the three primary G-protein transduction families: (C) Gi/o; (D) Gs; (E) Gq/11. (F) Adjacency matrices similar to A-E, except row and column factor vectors were calculated as means of CPM values across neuron types comprising indicated subclasses. (Cladograms and taxonomic color codes as delineated in *Figure 6—figure supplements 1–8*). Links below point to source data files and similar plots for all 37 VISp and ALM type and subclass adjacency matrices, and to additional quantitative analysis of coupling matrix hierarchies (*Figure 6—figure supplement 9*) and morphologies and correlations (*Figure 6—figure supplement 10*).

The online version of this article includes the following source data and figure supplement(s) for figure 6:

**Source data 1.** NP coupling predictions by cognate pair and type for area VISp (archive containing 37 CSV files, one for each of the 37 cognate pairs listed in *Table 3* and represented in *Figure 6—figure supplements 1–8*).
**Source data 2.** NP coupling predictions by cognate pair and type for area ALM (archive containing 37 CSV files, one for each of the 37 cognate pairs listed in *Table 3* and represented in *Figure 6—figure supplements 1–8*).
**Figure supplement 1.** Neurotaxonomic coupling predictions for cognate NPP/NP-GPCR pairs #1-#4.
**Figure supplement 2.** Neurotaxonomic coupling predictions for cognate NPP/NP-GPCR pairs #5-#9.
**Figure supplement 3.** Neurotaxonomic coupling predictions for cognate NPP/NP-GPCR pairs #10-#14.
**Figure supplement 4.** Neurotaxonomic coupling predictions for cognate NPP/NP-GPCR pairs #15-#18.
**Figure supplement 5.** Neurotaxonomic coupling predictions for cognate NPP/NP-GPCR pairs #19-#23.
**Figure supplement 6.** Neurotaxonomic coupling predictions for cognate NPP/NP-GPCR pairs #24-#27.
**Figure supplement 7.** Neurotaxonomic coupling predictions for cognate NPP/NP-GPCR pairs #28-#32.
**Figure supplement 8.** Neurotaxonomic coupling predictions for cognate NPP/NP-GPCR pairs #33-#37.
**Figure supplement 9.** Hierarchical relationships of coupling matrices.
**Figure supplement 10.** Coupling matrix localization and correlation metrics for VISp and ALM areas.

*Figure 6C-E* represents 8 more of the 37 cognate pair coupling matrices predicted for VISp. Along with *Figure 6A and B*, these exemplify the wide variety of neuron-type-specific coupling motifs resulting from transcriptomic prediction. Most coupling matrices (i.e., pairs 1, 9, 29), predict significant coupling over wide swaths of type-pairs, approaching 20% of the entire matrix. A few matrices at the other extreme, such as 6 and 34, predict very sparse coupling. Other predictions are intermediate in sparsity. As one might expect, similar patterns are evident in the downsampled, subclass level predictions of *Figure 6F*. Even from the small subset of the 37 coupling matrix plots shown in *Figure 6*, it is evident that both type-level and subclass-level matrices are densely tiled by predictions of connectivity. Inspection of *Figure 6* and similar plots for the remainder of the 37 cognate pairs (*Figure 6—figure supplements 1–8*) also reveals that there is a great deal of cross-network redundancy, with multiple pairs covering a large majority of the coupled types and subclasses, sometimes within and sometimes crossing Gα family boundaries. These observations will be strengthened by the analysis of *Figure 7* below.

Finally, *Figure 6* illustrates the tendency of coupling predictions from most cognate NP pairs to fall in contiguous 'patches' of the full coupling matrix. This is a natural reflection of the strong tendency of both NPP and NP-GPCR expression to align with early nodes in the Tasic 2018 hierarchical clustering which was also evident in *Figures 3* and *5*. The broadest example of coupling matrix patches reflecting hierarchical neurotaxonomy structure is provided by the observation that most sizable coupling patches fall strictly within single quadrants of glutamatergic-GABAergic neuron type pairing. Variations in coupling matrix structure across all 37 cognate NP pairs are represented in more quatitative terms by *Figure 6—figure supplements 9* and *10*. Additional details regarding the generation of the coupling matrices are provided in Materials and methods.

## Prediction of second-messenger impacts from neurotaxonomic NP gene expression profiles

For compact visualization of predicted signaling impacts of multiple distinct peptidergic networks and to facilitate empirical tests of such predictions based on calcium and cyclic AMP sensors (see Discussion), we developed the 'ISQ' graphic exemplified in *Figure 7*. This treatment makes use of the trichotomous G-protein primary transduction family approximation described in Introduction and

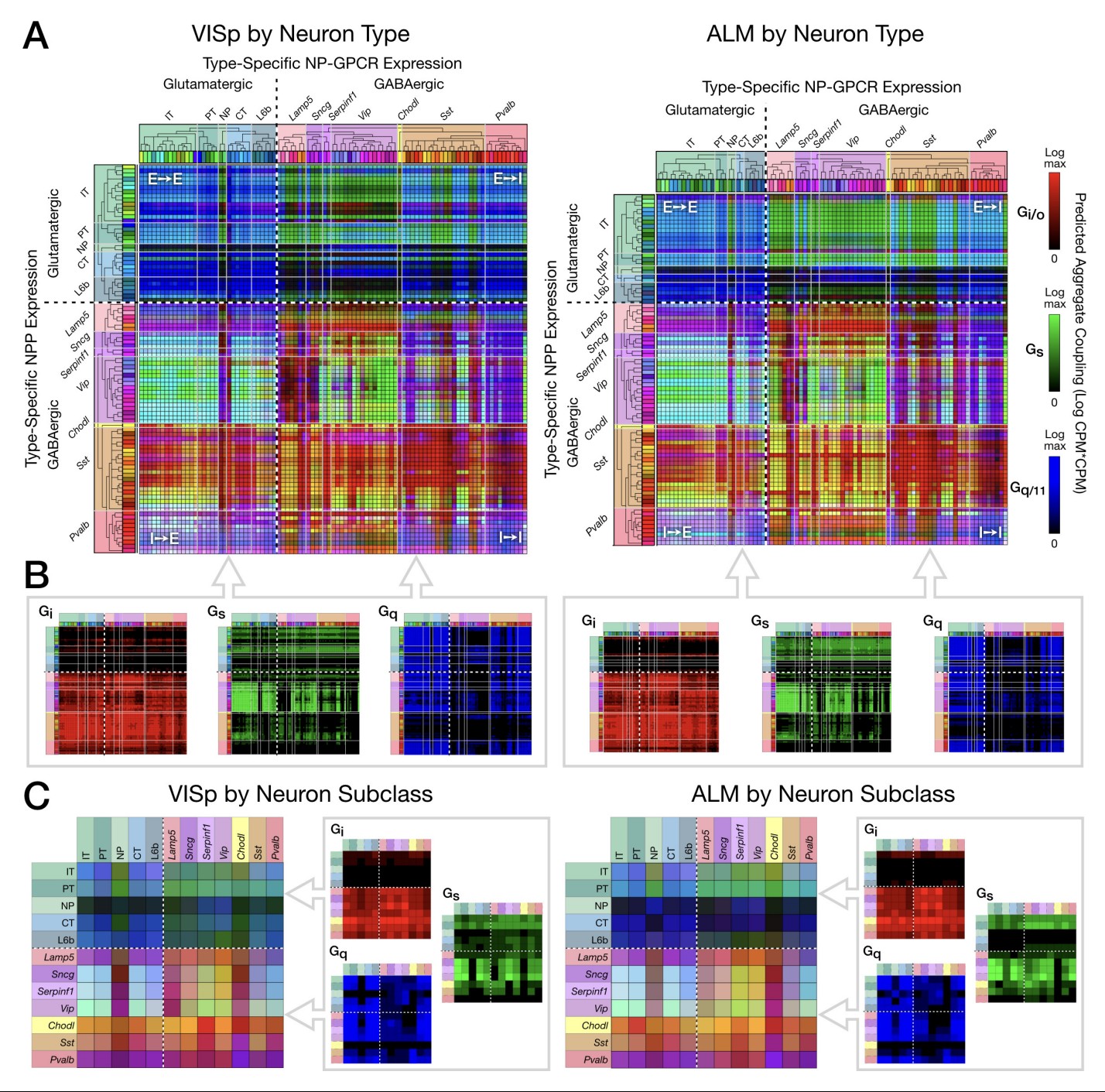

**Figure 7.** Pooling NP-GPCRs by primary Gα family enables neurotaxonomic prediction of primary NP-GPCR signaling impacts across 37 cognate NP pairs (see text) for each of areas VISp and ALM. (**A**) ISQ color maps representing coupling matrix predictions at the Tasic 2018 neurotaxonomy type level, merging Gi/o (red), Gs (green) and Gq/11 (blue) primary Gα family components. (**B**) Component primary Gα family (color) channels prior to merger displayed in (**A**). (**C**) Coupling matrix plots as in (**A**) and (**B**), except generated at the higher Tasic 2018 neurotaxonomy subclass level. Individual aggregate matrix components as in (**B**) are plotted at the right for both VISp and ALM. Dashed white crosses overlaying each matrix plot demarcate glutamatergic and GABAergic classes and the four corresponding matrix quadrants (E→ E, E→I, I→I and I→E) as in the individual matrix plots of *Figure 6*. (Cladograms and taxonomic color codes as delineated in *Figure 3—figure supplement 1*).

The online version of this article includes the following source data for figure 7:

**Source data 1.** VISp coupling predictions by Gα family and type.
**Source data 2.** ALM coupling predictions by Gα family and type.

delineated in *Table 3* above: the Gi/o family ('I') inhibits production of cyclic AMP, the Gs family ('S') stimulates production of cyclic AMP) and the Gq/11 family ('Q') augments intracellular calcium dynamics. Such trichotomy is certainly an oversimplification, as it is known that downstream GPCR signal transduction is richly multifaceted (*Weis and Kobilka, 2018*) and that some GPCRs may signal via members of multiple Gα families, but we postulate here that this simplified scheme may nonetheless offer a first approximation useful for the design of exploratory experimentation and theory.

*Figure 7* displays three-channel 'ISQ' (red, green, blue) color maps predicting coupling in areas VISp and ALM based on aggregation across three primary Gα families (Gi/o, Gs and Gq/11). Individual cognate-pair coupling matrices were computed as in *Figure 6*, log10 scaled, individually normalized to maximum values, then summed into a red, green or blue color channel by primary Gα family as listed in *Table 3*. *Figure 7A* merges red, green and blue (i.e., Gi/o, Gs, Gq/11) color channels for the Tasic 2018 neuron-type-level. *Figure 7B* displays the three component channels individually. The dashed white crosses on these and following coupling matrix both plots divide these ISQ maps into four E-I quadrants as in *Figure 6*. Major features of the ISQ maps are clearly very similar for VISp and ALM. *Figure 7C* show aggregated matrix plots generated in similar fashion for the subclass-level neurotaxonomy.

The ISQ maps of *Figure 7* exhibit a number of interesting features. (1) The aggregate matrices show that the 37 cognate pairs cumulatively predict coupling that densely tiles the entire neuron-type coupling matrix, with the largest area of relatively weak coupling being that from NP, CT and L6b subclasses to GABAergic neurons. (2) Aggregate predictions are highly conserved between VISp and ALM areas. (3) All four E-I quadrants show coupling representative of all three Gα families. (4) There is nonetheless some family predominance within each quadrant: Gi/o (blue) in the E→E quadrant, Gs (green) in the E→I quadrant, and Gi/o (red) in the I→I and I→E quadrants. (5) As is particularly notable in the component matrix plots, Gi/o (red) signaling is the most heavily concentrated, with quite little signaling expressed in the top two (E-E and E-I) quadrants but tiling the bottom two (I-E and I-I) quite thoroughly. Gq/11 signaling shows a weaker, but still noticeable tendency toward concentration in the two left quadrants (E-E and I-E). Gs signaling exhibits distinct zone of concentration, but these are not well captured by the quadrant structure. (6) The presence of cyan, yellow and purple (blended) colors in the merged matrix plots (A and C), particularly in the bottom quadrants is indicative of coincidence of signaling impacts of multiple Gα families at individual type (A) and subclass (C) intersections.

## Discussion

Light from single-cell transcriptomics is now beginning to illuminate dark corners of cellular neuroscience that have long resisted mechanistic and functional analysis (*Fan et al., 2018*; *Fishell and Kepecs, 2019*; *Földy et al., 2016*; *Gokce et al., 2016*; *Luo et al., 2018*; *Okaty et al., 2011*; *Paul et al., 2017*; *Shekhar et al., 2016*; *Tasic et al., 2018*; *Tasic et al., 2016*; *Telley et al., 2016*; *Zeng and Sanes, 2017*). Cortical neuropeptide signaling may be one such corner. While profound impacts of neuropeptide signaling are well-established in a wide range of non-mammalian and subcortical neural structures (*Borbély et al., 2013*; *Burbach, 2011*; *Elphick et al., 2018*; *Grimmelikhuijzen and Hauser, 2012*; *Katz and Lillvis, 2014*; *Jan et al., 1979*) and there certainly is an excellent literature on cortical neuropeptide signaling (*Crawley, 1985*; *Férézou et al., 2007*; *Gallopin et al., 2006*; *Gomtsian et al., 2018*; *Hamilton et al., 2013*; *Liu et al., 2018*; *Mena et al., 2013*; *Mitre et al., 2018*; *Owen et al., 2013*; *Rossier and Chapouthier, 1982*; *Williams and Zieglgänsberger, 1981*), published physiological results are surprisingly rare given the breadth of neuroscientific interest in cortex. The new transcriptomic data analyzed here suggest a possible explanation for this relative rarity. Though many NPP and cognate NP-GPCR genes are expressed abundantly in all or very nearly all neocortical neurons, such expression is highly differential, highly cell-type specific, and often redundant. These previously uncharted differential expression factors may have hindered repeatable experimentation. Our analysis supports this unwelcome proposition but may also point the way to more productive new perspectives on intracortical peptidergic neuromodulation.

## Summary of findings

The present single-cell analysis establishes that mRNA transcripts from one or more of 18 NPP genes are detectable in over 97% of mouse neocortical neurons (*Figure 1A,B*) and that transcripts of one or more of 29 cognate NP-GPCR genes are detectable in over 98% (*Figure 2A,B*). Transcripts of at least one of the 18 NPP genes are present in the vast majority of cortical neurons at extremely high copy number (*Table 1*), suggesting the likelihood of brisk translation into neuropeptide precursor proteins. Brisk synthesis of precursor proteins further suggests brisk processing to active neuropeptide products and secretion of these products. Likewise, NP-GPCR transcripts rank high in abundance compared to most other transcripts of protein-coding genes (*Table 2*), supporting the likelihood of functional receptor products. Our observations thus support the proposition that all, or very nearly all, neocortical neurons, both glutamatergic and GABAergic, are also both neuropeptidergic and modulated by neuropeptides. We are not aware of any previous empirical support for quite such a strong conclusion.

Leveraging the analytical power of the Tasic 2018 transcriptomic neurotaxonomy, we find that patterns of differential expression of the 18 NPP and 29 NP-GPCR genes are very highly specific to neuron types as discerned from genome-wide transcriptomic analysis (*Figure 3*). Though much additional work (e.g., see *Cadwell et al., 2017*; *Daigle et al., 2018*; *Moffitt et al., 2016*; *Shah et al., 2016*; *Wang et al., 2018*; *Zeng and Sanes, 2017*) will be needed to fully reconcile new transcriptomic neurotaxonomies such as the Tasic 2018 example with existing anatomical and physiological neurotaxonomies, it seems very likely that some such reconciliation will eventually take place, and that the dimensions of neurotaxonomy will be expanded to include emerging connectomic data (*Jonas and Kording, 2015*).

Our analysis shows that very intricate single-cell (*Figures 1B,C* and *2B,C*) and neurotaxonomic (*Figure 5*) patterns of expression of 18 NPP and 29 cognate NP-GPCR genes are very rigorously conserved between VISp and ALM, two distant and quite different areas of neocortex. Such strong conservation would seem improbable if these intricate patterns resulted from ephemeral factors such as local electrical activity or modulation status. Rather, we suggest that this strong conservation is more likely to reflect a really fundamental evolutionary and developmental connection between neuropeptide network architectures and adaptive cortical circuit function.

Following earlier indications that neurons may express multiple NPP genes, for example (*Mezey et al., 1999*), our analysis establishes that expression of multiple NPP genes in individual neurons may be the rule in cortex (*Figure 1C*). Our analysis also establishes the generality of expression of multiple NP-GPCR genes in individual cortical neurons (*Figure 2C*). The significance of these observations remains to be explored but should be viewed in light of recent discoveries of large numbers and great diversity of transcriptomic neuron types in neocortex and many other brain regions. Combinatorial expression of neuropeptide precursor and receptor genes obviously expands the prospects for molecular multiplexing that may allow selective communication amongst a multiplicity of distinct neuron types even though the signaling molecules propagate in diffuse paracrine fashion. It is also good to keep in mind, however, that the selectivity of NP-GPCRs for particular peptide moieties is not perfect. Various kinds of concentration-dependent 'crosstalk' between nominally separate peptidergic networks are therefore possible. Here in the interests of simplicity we have confined explicit peptidergic signaling predictions to the highest affinity pairings of NPP and NP-GPCR gene products (e.g., see *Alexander et al., 2017*).

We also find that a modest set of 47 neuropeptide-signaling genes permits transcriptomic neuron type classification that is exceptionally precise in comparison to other similarly small gene sets (*Figure 4*). Connections between neuronal cell-type differentiation and differential expression of neuropeptides were first recognized by the widespread use of neuropeptide immunoreactivity to discriminate interneuron types (*DeFelipe et al., 2013*). The exceptional power of neuropeptide genes as cell type markers is also evident in the Tasic 2018 neuron-type nomenclature (see *Tasic et al., 2018*) and bold red type highlights in *Figure 3—figure supplement 1*) and is noteworthy in other recent single-cell transcriptomic analyses of mouse neuron differentiation (*Huang and Paul, 2019*; *Paul et al., 2017*; *Sugino et al., 2019*; *Zeisel et al., 2018*). The tight alignment of neuron type classifications based solely on neuropeptide-signaling gene expression with the classifications based on genome-wide expression patterns, as evident in *Figure 4*, offers an intriguing suggestion of a very deep and fundamental connection between the expression of evolutionarily

ancient neuropeptide-signaling genes and the differentiation of neuron type identities during metazoan speciation.

## The structures of predicted neuropeptidergic modulation networks

Our analysis delineates neuron-type-specific expression of 37 cognate pairs amongst the 18 NPP and 29 NP-GPCR genes analyzed (*Table 3*). Each of these pairs can be taken to predict a modulatory connection from cells expressing a particular NPP gene, via a secreted NP product, to cells expressing the particular NP-GPCR gene (*Figure 6*). Each pair thus establishes the prospect of a directed modulatory network with nodes defined by the neurotaxonomic identities of the transmitting NPP-expressing and the receiving NP-GPCR-expressing neurons. The analyses represented in *Figures 1*, *2*, *3* and *5* and *Table 3* establish that at least one of the 37 pairs directly involves every neuron sampled, and that the vast majority of neurons are directly involved in more than one of the 37 predicted networks. The nearly complete adjacency matrix tiling evident in *Figures 6* and *7* remarkably suggests that at least one of the 18 peptides considered here may directly interconnect almost every cortical neuron type with almost every other neuron type. Because of this saturated, multiplexed coverage of all neurons and neuron types, we refer to these predicted neuropeptidergic networks as 'dense'.

Transcriptomic prediction of paracrine local signaling from GABAergic neuron sources is particularly compelling. Because few cortical GABAergic neurons have axons that project beyond the confines of a single cortical area, considerations of diffusion physics and the limited lifetime of peptides after secretion strongly imply that secreted neuropeptides act locally. On the other hand, most of the glutamatergic neurons do emit long axons, so it is possible that neuropeptides secreted from such neurons may act in remote cortical or extracortical projection target areas. Even so, most cortical glutamatergic neurons also have locally ramifying axon branches and may also secrete neuropeptides from their local dendritic arbors (*Vila-Porcile et al., 2009*). The high cortical expression of NP-GPCRs cognate to NPP genes expressed by glutamatergic neurons in the same local area suggests a scenario supportive of local modulatory signaling from glutamatergic neuron sources, though this case may not be quite as strong as that for strictly local GABAergic neurons. That said, the much more profuse expression of NPP genes in GABAergic neuron types along with the somewhat more profuse NP-GPCR expression in glutamatergic types does suggest a 'prevailing wind' of peptidergic signaling, blowing mainly from GABAergic to glutamatergic neurons, as presaged in an earlier microarray analysis of developing mouse cortex (*Batista-Brito et al., 2008*).

Though our NP network predictions are entirely consistent with decades of pioneering work on peptidergic neuromodulation and cortical gene expression (*Burbach, 2011*; *Hökfelt et al., 2013*; *van den Pol, 2012*), it is only with the recent advent of single-cell and neurotaxonomics methods that such specific predictions have become possible and, most importantly, testable.

## Testing peptidergic network predictions

The present predictions regrading cortical neuropeptidergic coupling are based on detection of cellular mRNA transcripts, but prediction from such data depends upon (1) extrapolation from cellular mRNA census to inference about rates of synthesis, processing, localization and functional status of cellular NPP and NP-GPCR proteins, (2) assumptions about neuropeptide diffusion and lifetime in cortical interstitial spaces, (3) assumptions about signaling consequences of neuropeptide binding to cortical NP-GPCR receptors. Though we have already discussed several factors that mitigate such concerns, we stipulate here that these uncertainties remain substantial. Nonetheless, we expact that these same uncertainties will define paths for very productive future research.

Physiological and anatomical experimentation will be essential to testing transcriptomic predictions of intracortical neuropeptide signaling. We have suggested that such work may have been frustrated in the past by irreproducibility due to the uncharted multiplicity, neuron-type-specificity, and redundancy of NPP and NP-GPCR expression. This conundrum may now be resolved with the emergence of transcriptomic neurotaxonomies and new tools for experimental access to specific cortical neuron types. Such access may be either *prospective*, using Cre and/or Flp driver lines (*Daigle et al., 2018*; *He et al., 2016*; *Madisen et al., 2015*) or viral vectors (*Dimidschstein et al., 2016*) of substantial neuron-type-specificity, or *retrospective* by multiplexed FISH (*Lein et al., 2017*; *Zeng and Sanes, 2017*), immunostaining (*He et al., 2016*; *Xu et al., 2010*), patch-seq

(*Cadwell et al., 2017*; *Lein et al., 2017*) or morphological classification methods (*DeFelipe et al., 2013*; *Zeng and Sanes, 2017*). These and other new molecular tools like those discussed below now seem poised enable truly decisive and repeatable tests of neuron-type-specific transcriptomic predictions of peptidergic signaling. It will be critical, however, for the field to have continually updated access to rapidly growing bodies of genetic and transcriptomic data and to the requisite animal strains and labeling materials.

A vast pharmacopeia of well-characterized specific ligands and antagonists for most NP-GPCRs (*Alexander et al., 2017*) will be bedrock for the functional analysis of neuron-type-specific peptide signaling. For analysis of type-specific neuropeptide signaling in network context (i.e., ex vivo slices and in vivo), newer optophysiological methods of calcium imaging and optogenetic stimulation/inhibition will certainly join electrophysiology as foundations for measurement of neuropeptide impacts. In addition, many new tools more specific to neuropeptide signaling are emerging. Super-resolution 3D immunohistologies like array tomography (*Smith, 2018*) and 3D single-molecule methods (*Jia et al., 2014*; *von Diezmann et al., 2017*) will enable imaging of dense-core vesicle localization and neuropeptide contents in type-specific network anatomical context. Genetically encoded fluorescent dense-core vesicle cargos will allow real-time detection of neuropeptide secretion (*Ding et al., 2019*), while genetically encoded sensors of extracellular GPCR ligands (*Patriarchi et al., 2018*; *Sun et al., 2018*), GPCR activation (*Haider et al., 2019*; *Hill and Watson, 2018*; *Livingston et al., 2018*; *Ratnayake et al., 2017*; *Stoeber et al., 2018*), G-protein mobilization (*Ratnayake et al., 2017*), cAMP concentration (*Hackley et al., 2018*; *Ma et al., 2018*), protein kinase activation (*Chen et al., 2014*) and protein phosphorylation (*Haider et al., 2019*) will enable fine dissection of NP dynamics and NP-GPCR signal transduction events (*Spangler and Bruchas, 2017*). In addition, new caged NP-GPCR ligands (*Banghart et al., 2018*) and antagonists (*Banghart et al., 2013*) will provide for precise spatial and temporal control for NP receptor activation. All of these tools have already been proved at least in principle, and all should be readily applicable to testing specific hypotheses derived from the type-specific peptidergic signaling predictions set forth here (*Figures 6* and *7* and their supplements).

## Prospects for elucidating cortical homeostasis, modulation and plasticity

Our results suggest that densely multiplexed peptidergic networks could play very significant roles in the homeostasis, modulation and plasticity of cortical synaptic networks. Due to the clearly formidable complexity of cortical networks, however, a real grasp of the myriad network interactions implicated is certain to require theoretical and computational approaches, in addition to experimental biophysics tests as outlined in the preceding section. Work at the fertile intersection of the neuroscience and the computer science of learning (*Dayan and Abbott, 2001*; *Huh and Sejnowski, 2017*; *Koch and Segev, 1998*; *Lillicrap et al., 2016*; *Marblestone et al., 2016*; *Guerguiev et al., 2017*; *Song et al., 2000*) seems particularly relevant to fathoming the possible significance of the neuropeptidergic networks we predict here.

Neuroscience and computer science efforts to model or engineer adaptive neural networks (be they biological or artificial) share the hard problem of optimally individualized adjustment of very large numbers of what both fields know as 'synaptic weights'. At the heart of this challenge is 'credit assignment', that is, the assignment of 'credit' (or 'blame') to guide the strengthening (or weakening) of the small subset of synapses that actually contribute differentially to success (or failure) in a given perceptual, mnemonic or motor task. Neuroscientists struggle with the credit assignment problem as they search for biological learning rules. Computer scientists are driven by a quest for greater computational efficiency in training artificial networks and the prospect that evolution may have developed superior strategies. One concept that has come into prominence as a candidate biologically plausible solution to the credit assignment problem is that of modulated 'Hebbian' or 'spike-timing-dependent' plasticity (STDP) (*Bengio et al., 2016*; *Dan and Poo, 2006*; *Farries and Fairhall, 2007*; *Florian, 2007*; *Frémaux and Gerstner, 2015*; *Froemke, 2015*; *Izhikevich, 2007*; *Marblestone et al., 2016*; *Pawlak et al., 2010*; *Poo et al., 2016*; *Roelfsema and Holtmaat, 2018*; *Xie and Seung, 2003*) While most biological studies of modulated STDP so far have focused on the monoamine neuromodulator dopamine (*Brzosko et al., 2019*; *Izhikevich, 2007*; *Kuśmierz et al., 2017*; *Schultz, 2015*) known commonalities of signal transduction downstream from widely varying GPCRs suggest that NP-GPCRs could play roles in credit assignment analogous to those postulated

for dopamine-selective GPCRs (*Hamilton et al., 2013*; *Roelfsema and Holtmaat, 2018*; but see *Edelmann and Lessmann, 2011*).

Deeper understanding of neuromodulation roles in adaptive cortical function seems certain to require a framework for integrating consideration of the panoply of possible activity-dependent modulatory networks with modulated excitatory and inhibitory synaptic networks. *Figure 8* conceptualizes one such framework schematically, using a common neurotaxonomy to integrate statistics of multiple neuromodulatory and multiple synaptic signaling networks. Panels A-C idealize a logic for prediction of neuropeptidergic connectivity statistics from transcriptomic data. Panel D cartoons the use of a common neurotaxonomy to integrate probabilistic NP network graphs (panels B,C; three in this case) and multiple synaptic networks (panels E,F; two in this case) into a single graph representing superimposed modulatory and synaptic network. The present analysis suggests that a more realistic materialization of the *Figure 8* schematic would involve approximately 100 neuron types and dozens of NPP and NP-GPCR genes. It would also require information that is presently unavailable about excitatory and inhibitory synaptic connectivity statistics in such a neurotaxonomic framework. It is very encouraging, however, that vigorous ongoing efforts (e.g., see *Daigle et al., 2018*; *Jonas and Kording, 2015*; *Swanson and Lichtman, 2016*; *Tasic, 2018*; *Zeng and Sanes, 2017*) suggest that such information is on the way. A view of cortical circuitry as a superimposition of multiple modulatory and synaptic networks, linked by a common neurotaxonomy as idealized in *Figure 8*, may prove essential to fathoming the interplay of slow neuromodulation and fast synaptic signaling necessary for adaptive cortical function.

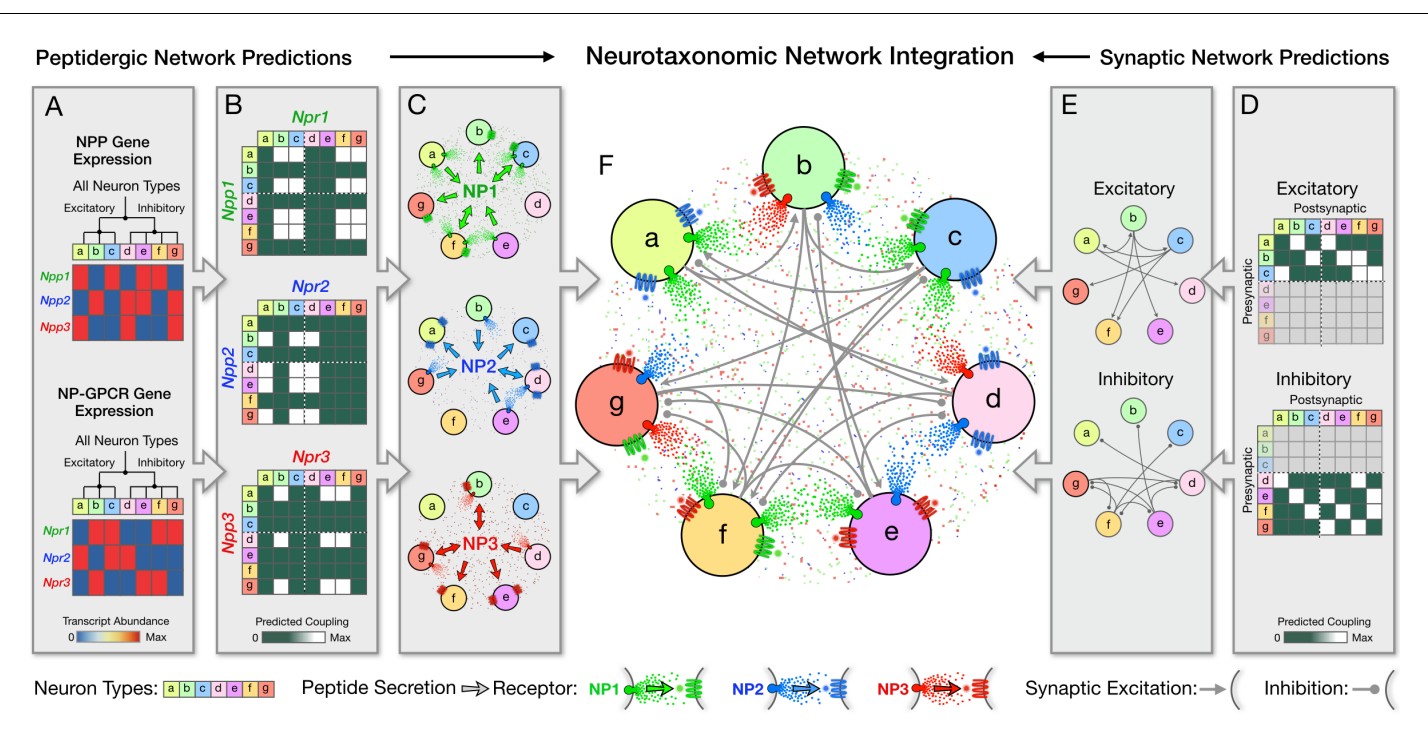

**Figure 8.** Neurotaxonomy offers a framework for integrating statistical descriptions of multiple modulatory and synaptic networks, as schematized here for purely fictitious neuron types, transcriptomic and connectomic data. Multiple directed network graphs are predicted here for peptidergic networks from transcriptomic data (A–C) and for synaptic networks from connectomic data (D,E) to predict a modulated synaptic network (F). (A) Transcriptional heat maps representing expression of three fictitious NPP genes (Npp1,2,3) and three cognate NP-GPCR genes (Npr1,2,3) across a neurotaxonomy comprising seven fictitious neuron types (a-c excitatory; d-g inhibitory). (B) Adjacency matrices derived from expression data in (A) as outer products of column and row factor vectors representing NPP and NP-GPCR expression, respectively. (C) Directed network graphs representing the same three NP networks, diagramming paracrine coupling by three neuropeptides (NP1,2,3) with routing of broadcast diffusive signals determined by differential expression of peptide-receptor pairings. (D) Neurotaxonomic adjacency matrices expressing excitatory and inhibitory synaptic connection statistics. (E) Synaptic network graphs derived from (D). (F) Directed multigraph illustrating use neurotaxonomy to integrate the three modulatory graphs and two synaptic connectivity graphs.

## Implications for psychopharmacology

Neuropeptidergic signaling molecules have long beguiled as potential neuropsychiatric drug targets (*Hökfelt et al., 2003*; *Hoyer and Bartfai, 2012*). There seems to be some disappointment, however, in the returns on what has been rumoured to be very large research investments. The present study raises the possibility that both NP-targeted drug discovery and the reproducibility of physiological experimentation have been hindered by the same uncharted multiplicity, cell-type-specificity and redundancy of NPP and NP-GPCR expression. By charting these waters, single-neuron transcriptomic analysis may improve the odds substantially for both reproducible research and psychiatric drug development.

Today's psychiatric pharmaceuticals almost all target signaling by the monoamine neuromodulators dopamine, serotonin, noradrenaline and/or histamine and their selective GPCR receptors (*Data-Franco et al., 2017*; *Hamon and Blier, 2013*; *Millan et al., 2015*; *Urs et al., 2014*). Because they are so numerous, neuropeptide signaling systems may be much more neuron-type specific than monoamines. Greater neuron-type-specificity may translate to NP-targeting drugs being less troubled by side-effects and compensation (*Hoyer and Bartfai, 2012*). Moreover, while GPCRs have long been known as among the most 'druggable' of targets (*Gurrath, 2001*; *Lundstrom, 2009*), the 'druggability' of GPCRs is currently advancing very rapidly due to advances in GPCR structural biology and molecular dynamic simulations (*Hilger et al., 2018*; *Koehl et al., 2018*; *Weis and Kobilka, 2018*). It seems likely that new knowledge of the neuron-type-specificity of NP signaling gene expression will substantially advance the development of NP-targeting pharmaceuticals.

## Conclusions

Because single-cell RNA-seq data enable prediction of complete protein parts lists of individual neurons, they open powerful new perspectives on neuronal differentiation, function and network architectures. The power of these new perspectives has been further enhanced by parallel development of transcriptomic neurotaxonomies. Here we have exploited both a pioneering large-scale RNA-seq dataset and its data-driven neurotaxonomy to pursue a new perspective on local neuropeptidergic modulatory signaling in mouse cortex. This work has revealed a surprisingly highly structured and abundant expression of cortical NPP and NP-GPCR genes: dozens of neuropeptide signaling genes are expressed at very high levels in very distinctive and highly conserved patterns. While entirely consistent with previous bulk transcriptomic and proteomic observations, it is only with the advent of the RNA-seq combination of single-cell resolution with genomic depth that this extreme structure and abundance has come into focus. We have endeavored here to shape these findings into specific and testable peptidergic signaling predictions in the hopes of guiding fruitful experimentation based on emerging transcriptomic neurotaxonomies, new means for genetic access to specific neuron types and powerful new tools for biophysical analysis of neuropeptide actions. The observations presented here suggest the intriguing possibility that the homeostasis, modulation and plasticity of cortical circuitry may involve local neuropeptidergic signaling networks of previously unrecognized abundance and density.

## Materials and methods

### Data and software resources

The present study is based on analysis of a resource single-cell mRNA-seq dataset acquired at the Allen Institute (*Tasic et al., 2018*) and available for download at http://celltypes.brain-map.org/rna-seq/ These RNA-seq data were acquired from a total of 22,439 isolated neurons, with detection of transcripts from a median of 9462 genes per cell  (min = 1,445; max = 15,338) and an overall total of 21,931 protein-coding genes detected. Neurons were sampled from two distant and very different neocortical areas: 13,491 neurons from primary visual cortex (VISp), and 8948 neurons from anterior lateral motor cortex (ALM). Tasic et al., harvested tissue specimens from a variety of transgenic mice expressing fluorescent proteins to enable enrichment of samples for neurons and for relatively rare neuron types by FACS sorting after dissociation. This enrichment procedure resulted, by design, in a disproportionate representation of GABAergic neurons, canonically ~20% of neurons (*Sahara et al., 2012*), such that the sampled neuron population is roughly half GABAergic (47%) and half glutamatergic (53%). The resource publication (*Tasic et al., 2018*) should be consulted for full details of

neuronal sample and library preparation, sequencing and data processing. Source data, spread-sheets, R scripts and other code used to generate all tables and figures presented here are available at https://github.com/AllenInstitute/PeptidergicNetworks (*Gala, 2019*; copy archived at https://github.com/elifesciences-publications/PeptidergicNetworks). A derived data set *np_gpcr_cpm.csv* was used for analyses summarizing CPM, region and metadata for the NPP and NP-GPCR genes. The primary Tasic 2018 data tables are available for download at http://celltypes.brain-map.org/rna-seq/.

## Data metrics

The Tasic 2018 single-cell RNA-seq data tables report the abundance of transcripts from individual neurons in both 'counts per million reads' (CPM) and 'fragments per kilobase of exon per million reads mapped' (FPKM) units. Our analysis of this data compares gene expression levels quantitatively, with two distinct use cases: (1) comparisons across large sets of different genes, and (2) comparisons of the same gene across different individual cells, cell types and brain areas. We have relied upon FPKM data (*Mortazavi et al., 2008*; *Pimentel, 2014*), for use case 1 (i.e., the *Tables 1* and *2* comparisons across genes). For use case 2 (as in all figures below), we have preferred the CPM units, because these units were used to generate the Tasic 2018 neurotaxonomy. While choices between CPM and FPKM units here should have little impact upon outcomes, it would seem inconsistent to use FPKM units to compare across cell types discerned on the basis of CPM units.

The NP signaling genes upon which the present analysis focuses are expressed very differentially across the sampled populations of individual mouse cortical neurons. That is, each gene is expressed at a high level in some subset of cells but at zero or very low levels in the remainder of the population. To compactly characterize such expression, we developed a 'Peak FPKM' (pFPKM) metric. This metric is generated by ranking single-cell FPKM values for a given gene across the entire population of 22,439 neurons sampled, then designating the FPKM value at the ascending 99.9[th] percentile point as pFPKM. This metric was designed to minimize effects of sporadic outliers and sample size while still closely approximating the actual peak expression value in even very small subsets of neurons expressing the gene in question. *Figures 1A* and *2A*, and their Source data files provide very detailed additional information about the single-cell RNA-seq value distributions sampled by the pFPKM metrics.

## Selection of the 18 NPP gene set

As noted in Introduction, usage and definitions of the term 'neuropeptide' vary widely across the current literature. It therefore seems unwise at present to claim that any attempted consensus list would accurately circumscribe all neuropeptides. For the purposes of the present work, we have relied therefore on the reasonably exhaustive list of 96 classical and candidate human and mouse NPP genes put forth in a widely cited publication (*Burbach, 2010*) and related website (http://neuropeptides.nl/, last accessed 10 October 2019). To reconcile this list to current mouse gene nomenclature, we used both the HGNC nomenclature ((https://www.genenames.org/, last accessed 10 October 2019) and the Mouse Genome Database (MGD) (http://www.informatics.jax.org, last accessed 10 October 2019). The result is the list of 94 putative mouse NPP genes presented in *Supplementary file 3*, which also tabulates the pFPKM values and percentile scores compiled for each NPP genes from the Tasic 2018 dataset. These 94 NPP genes were further segregated using a preliminary (early 2018) version of the Tasic 2018 neurotaxonomy to select NPP genes exhibiting median CPM expression levels > 10 in one or more neuron type in VISp and ALM cortex. This screening resulted in the list of 39 such NPP genes represented in *Supplementary file 4*, with most exceeding the 10 CPM threshold by a large margin (observed range was 24–4100 CPM). *Supplementary file 4* also tabulates criteria that drove inclusion of only the 18 NPP genes represented in *Table 1* while 21 other cortically expressed NPP genes were excluded. The 18 select NPP genes include all but two (*Edn3* and *Gal*) genes for which transcripts ranked in the top quintile by pFPKM of the 94 putative NPP genes as tabulated in *Supplementary file 3*.

## Selection of the 29 NP-GPCR gene set

The 18 select NPP genes listed in *Table 1* were used to search manually for cognate NP-GPCRs expressed in mouse cortex, relying primarily on ligand/receptor pairing data retrieved from the

IUPHAR/BPS Pharmacology website (http://www.guidetopharmacology.org, accessed in March, 2018) and the Tasic 2018 NP-GPCR expression data tabulated in *Supplementary file 5*. This process resulted in selection of the 29 mouse NP-GPCR genes listed in *Table 2*, which also lists for each the corresponding cognate NPP gene or genes used to root the search. The matching of NP-GPCR and NPP genes in *Table 2* neglects a few receptor/ligand pairings rated on the IUPHAR/BPS website as very low in affinity compared to primary pairings.

## Autoencoder-based classifier development and evaluation methods

### Gene sets

Table of different sets of genes used for experiments shown in *Figure 4*:

| Gene set | Description |
| --- | --- |
| NP47 | The combined set of 18 NPPs and 29 NP-GPCRs |
| HE | 6083 genes selected based on maximum value across all neurons in the dataset |
| DE | 4020 differentially expressed genes for Tasic 2018 neurotaxonomy |
| DE47 | 47 most variable genes selected from the set of DE genes |
| Rand47 | Random subsets of 47 genes drawn from the set of HE genes |
| Rand47 ExpMatched | Random subsets of 47 genes such that the maximum expression value approximately matches that of the NP genes |

### Autoencoder network architecture

Autoencoders are multi-layer, feedforward neural network models that consist of encoder/decoder subnetworks. In its basic realization, the encoder subnetwork learns to compress the high dimensional input into a low dimensional representation, from which the decoder subnetwork estimates the original input. We constructed a network with two autoencoders, with eight hidden layers each. The architecture of the first autoencoder (HE Genes autoencoder, *Figure 4C*) is Input(6083) → Dropout(0.8) → Dense(100) → Dense(100) → Dense(100) → Dense(100) → Dense(d) → Batch Normalization (latent representation $z_1$) → Dense(100) → Dense(100) → Dense(100) → Dense(100) → Dense (6083), and the architecture of the second autoencoder (NP Genes autoencoder, *Figure 4B*) is Input (47) → Dropout(0) → Dense(x) → Dense(x) → Dense(x) → Dense(x) → Dense(d) → Batch Normalization (latent representation $z_2$) → Dense(x) → Dense(x) → Dense(x) → Dense(x) → Dense(47). The numbers in parentheses of Dense denote the number of fully connected units in those layers. All Dense layer units use the rectified linear (ReLU) function as the nonlinear transformation except for those in the Dense(d) layers, which do not use a nonlinear transformation. For results using the NP genes autoencoder x = 50; tests with x = 25, led to qualitatively similar results (not shown) and did not change overall conclusions of the analyses. The Dropout layer (*Srivastava, 2014*) is used with dropout probability = 0.8 to regularize the HE Genes autoencoder and prevent over-fitting. The numbers of input/output units in each network match the number of input genes. The two dimensional representations (d = 2) shown in *Figure 4A–B*, and the five dimensional (d = 5) representations used in *Figure 4D* are the outputs of the Batch Normalization layer (*Ioffe and Christian, 2015*) for the respective networks. We determined the optimal latent space dimensionality d = 5 for the quantitative analysis by varying the latent space dimensionality of the HE Genes network between 2 and 20 dimensions and choosing the value that maximized the QDA analysis-based cell type classification accuracy for the HE genes (see *Figure 4—figure supplement 2*).

### Autoencoder training

Both autoencoder networks were trained using the backpropagation algorithm with the Adam optimizer (*Kingma and Ba, 2014*) and a batch size of 956. The HE genes autoencoder was trained for 50,000 epochs using the mean squared error between the input and the output layers as the loss function. The NP genes autoencoder was trained for 10,000 epochs using L = R+λC as the loss function, where R denotes the mean squared reconstruction loss as in the HE genes network, C denotes the penalty for mismatch between the latent representations, and λ = 100 is the weighting scalar

between the two terms. After training the HE genes network and obtaining the latent representation $z_1$ for each cell, C calculates the mean squared error between the latent representation of the NP genes network $z_2$ and $z_1$, while simultaneously normalizing variance along the narrowest direction for $z_2$. The two additive loss terms, R and C, together minimize the reconstruction error while attempting to match the representation learned using only the HE gene set. The same procedure was used for all small gene subsets including NP and random gene sets. Python implementations of the networks using the Tensorflow and Keras libraries are included in the code repository.

## Quantifying abilities of gene sets to classify cell types

The neurotaxonomy of *Tasic et al. (2018)* defines hierarchical relationships of neuronal cell types. For each gene set, we used Quadratic Discriminant Analysis (QDA) to train multiple classifiers on the latent space representations to predict labels at different levels of the cell type hierarchy. The different levels (nodes) in the hierarchy were characterized in Tasic 2018 with a resolution index measure. Here we re-normalized that resolution index measure to have a value of 0.0 for the class of neurons (root node), and 1.0 for the 115 VISp+ALM cell types (leaf nodes, inset in *Figure 4D*). All intermediate nodes in the hierarchical classification tree have a positive resolution index that is less than 1.0. We used this property of nodes in the hierarchical classification tree to assign a resolution index (RI) value to each cell. The procedure starts with a classifier that was trained using all the leaf node labels, that is all the 115 VISp+ALM cell type labels. Test cells that are classified correctly at this level are assigned RI = 1.0, which corresponds to the resolution index measure of the leaf nodes. Test cells that are incorrectly classified at this level of detail are re-assigned labels by a classifier that was trained on successively merged labels along the hierarchical tree till they are correctly classified. These cells receive the resolution index value of the node for which they are assigned the correct label. This procedure was performed using 13 fold cross validation for all the different gene sets, and the results were pooled.

## Peptidergic coupling matrices

For a given cortical area $A\{ALM, VIS_P\}$ , we denote by $NPP_A(g,t)$ the mean CPM expression matrix having entries NPP gene g and cell type *t*. Similarly, $NPGPCR_A(h,t)$ has as entries the expression of NP-GPCR gene *h* in type *t*. The *coupling matrix* $C^A_{(g,t)}$ of the pair (g,h) in area *A* is then defined $C^a_{(g,t)}(t,s) = \log_{10}(NPP_A(g,t) \times NPGPCR_A(h,s))$ for the fixed pair (g,h) in (NPP, NP-GPCR) as *t,s* range over all cell types in *A*. Matrices $C^A_{(g,t)}$ are formally the (square matrix) outer product $NPP_A \quad NPGPCR_A$ then presented in log10 units. Pooled representations are computed by averaging values of coupling matrices $C^A_{(g,t)}$ over 12 major cell types prior to rendering.

## Transduction mode predictions

Peptidergic coupling matrices are summed, log10 scaled and maximum normalized independently according to Gi/o, Gs and Gq/11 family membership, then displayed in red, green and blue, respectively. Pooled representations are computed by averaging type-level data over subclasses before similar rendering.

## Acknowledgements

We wish to thank the Allen Institute for Brain Science founder, Paul G Allen, for his vision, encouragement and support. This work was supported in part by award number R01NS092474 from the Office of the Director of National Institutes of Health and award number R01MH104227 from the National Institute of Mental Health. The content is solely the responsibility of the authors and does not necessarily represent official views of the National Institutes of Health.

# Additional information

## Funding

| Funder | Grant reference number | Author |
|---|---|---|
| National Institutes of Health | R01NS092474 | Stephen J Smith |
| National Institutes of Health | R01MH104227 | Stephen J Smith |
| National Institutes of Health | 1U24NS109113 | Stephen J Smith |

The funders had no role in study design, data collection and interpretation, or the decision to submit the work for publication.

## Author contributions

Stephen J Smith, Conceptualization, Data curation, Formal analysis, Supervision, Investigation, Visualization, Writing—original draft, Writing—review and editing; Uygar Sümbül, Conceptualization, Formal analysis, Supervision, Investigation, Visualization, Writing—original draft, Writing—review and editing; Lucas T Graybuck, Resources, Data curation, Formal analysis, Investigation, Visualization, Methodology; Forrest Collman, Conceptualization, Software, Formal analysis, Investigation, Visualization, Writing—original draft, Writing—review and editing; Sharmishtaa Seshamani, Resources, Data curation, Software, Formal analysis, Investigation, Methodology; Rohan Gala, Data curation, Software, Formal analysis, Investigation, Visualization, Writing—review and editing; Olga Gliko, Data curation, Software, Formal analysis, Investigation, Writing—original draft; Leila Elabbady, Data curation, Software, Formal analysis, Investigation, Visualization, Writing—original draft; Jeremy A Miller, Trygve E Bakken, Data curation, Formal analysis, Investigation, Visualization, Methodology; Jean Rossier, Data curation, Software, Investigation, Methodology, Writing—review and editing; Zizhen Yao, Resources, Data curation, Formal analysis, Investigation, Methodology; Ed Lein, Conceptualization, Resources, Supervision, Investigation, Methodology; Hongkui Zeng, Conceptualization, Resources, Investigation; Bosiljka Tasic, Resources, Supervision, Investigation, Methodology, Writing—review and editing; Michael Hawrylycz, Conceptualization, Data curation, Software, Formal analysis, Supervision, Investigation, Visualization, Methodology, Writing—original draft, Writing—review and editing

## Author ORCIDs

Stephen J Smith (iD) https://orcid.org/0000-0002-2290-8701
Uygar Sümbül (iD) http://orcid.org/0000-0001-7134-8897
Lucas T Graybuck (iD) http://orcid.org/0000-0002-8814-6818
Forrest Collman (iD) https://orcid.org/0000-0002-0280-7022
Rohan Gala (iD) https://orcid.org/0000-0003-1872-0957
Olga Gliko (iD) https://orcid.org/0000-0002-5014-7209
Leila Elabbady (iD) https://orcid.org/0000-0002-1452-1603
Jeremy A Miller (iD) http://orcid.org/0000-0003-4549-588X
Trygve E Bakken (iD) http://orcid.org/0000-0003-3373-7386
Jean Rossier (iD) https://orcid.org/0000-0003-1821-2135
Ed Lein (iD) http://orcid.org/0000-0001-9012-6552
Hongkui Zeng (iD) http://orcid.org/0000-0002-0326-5878
Bosiljka Tasic (iD) http://orcid.org/0000-0002-6861-4506
Michael Hawrylycz (iD) https://orcid.org/0000-0002-5741-8024

## Decision letter and Author response

Decision letter https://doi.org/10.7554/eLife.47889.sa1
Author response https://doi.org/10.7554/eLife.47889.sa2

## Additional files

### Supplementary files

• Supplementary file 1. Gene ontology results for 18 NPP genes.

• Supplementary file 2. Gene ontology results for 29 NP-GPCR genes.

• Supplementary file 3. pFPKM expression data for 94 putative mouse NPP genes.

• Supplementary file 4. Criteria for selecting 18 genes from a set of 39 cortically expressed NPP genes.

• Supplementary file 5. pFPKM expression data for 84 putative mouse NP-GPCR genes.

• Transparent reporting form

### Data availability

The present study is an analysis of a large transcriptomic dataset that is now freely available for download in its entirety at http://celltypes.brain-map.org/rnaseq/ and is described fully in a rigorously peer-reviewed publication (Tasic, et al., Nature 563:72-78, 2018). All code and intermediate data products involved in preparing this manuscript are freely available from a well-documented GitHub repository: https://github.com/AllenInstitute/PeptidergicNetworks (copy archived at https://github.com/elifesciences-publications/PeptidergicNetworks).

The following previously published dataset was used:

| Author(s) | Year | Dataset title | Dataset URL | Database and Identifier |
|---|---|---|---|---|
| Tasic B, Yao Z, Smith KA, Graybuck L, Nguyen TN, Bertagnolli D, Goldy J, Garren E, Economo MN, Viswanathan S, Penn O, Bakken T, Menon V, Miller JA, Fong O, Hirokawa KE, Lathia K, Rimorin C, Tieu M, Larsen R, Casper T, Barkan E, Kroll M, Parry S, Shapovalova N V, Hirchstein D, Pendergraft J, Kim TK, Szafer A, Dee N, Groblewski P, Wickersham I, Cetin A, Harris JA, Levi BP, Sunkin SM, Madisen L, Daigle TL, Looger L, Bernard A, Phillips J, Lein E, Hawrylycz M, Svoboda K, Jones AR, Koch C, Zeng H | 2018 | Gene-level (exonic and intronic) read count values for all mouse VISp and ALM samples | http://celltypes.brain-map.org/rnaseq/ | Allen Brain Map, rnaseq |

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
