## [Decision Letter]

**Acceptance summary:**

The impressive study by Smith and colleagues tackles in unprecedented fashion the relationship between the expression of neuropeptides and their cognate GPCRs. Using the same two cortical regions that the Allen institute has previously used for comparison (the visual and ALM cortices), Smith and colleagues compare the cell type specificity of peptide and receptors across the cortex. A number of fundamental observations are made: 1) virtually every neuronal type expresses multiple discrete types of NPP and associated receptors; 2) GABAergic cells show more NPP diversity while; 3) Glutamatergic cells show more diversity in receptor expression; 4) the 47 pairs of peptides and receptors can uniquely define cell types with high precision; 5) the relationships between peptides and receptors in stereotyped in a region specific manner. These are all observations of first rate importance, and I'd like to congratulate the authors for taking on a complex problem and discussing the underlying logic so systematically.

**Decision letter after peer review:**

Thank you for submitting your article "Single-cell transcriptomic evidence for dense intracortical neuropeptide networks" for consideration by *eLife*. Your article has been reviewed by three peer reviewers, and the evaluation has been overseen by a Reviewing Editor and Eve Marder as the Senior Editor. The following individuals involved in review of your submission have agreed to reveal their identity: Gordon Fishell (Reviewer #1); Bernardo L Sabatini (Reviewer #2); Matthew Ryan Banghart (Reviewer #3).

The reviewers have discussed the reviews with one another and the Reviewing Editor has drafted this decision to help you prepare a revised submission.

Summary:

Smith et al. performed analyses on a publicly available single-cell transcriptomic dataset from Tasic et al., 2018, to generate testable hypotheses regarding local neuropeptide signaling between neuronal cell types in the mouse cortex. Their main findings are: (1) 18 neuropeptides (NPs) and their receptors are highly expressed in most neurons across cortical areas, (2) most cells have multiple NPs and NP receptors, and (3) 18 NPs and their receptors are differentially expressed between neuronal cell types. These results highlight the importance of elucidating local neuropeptide signaling and their consequences on circuit and network activity. The study also provides a potentially useful framework for predicting intercellular signaling networks and generating experimental hypotheses from transcriptomic datasets. The study is timely, the manuscript is well written, and the results are interesting. For the revision, while no wet lab experiments are requested, the reviewers agree that additional data analysis, methodological explanations and discussion points, as described below, are warranted.

Essential revisions:

1) Table 1: Since the dataset is enriched for certain cell types including *Vip, Sst* and Ndnf (partly *Npy*) cells, it is not surprising that these peptides appear high up in the list. The FACS sorting process probably even selects cells with higher peptide expression within these subtypes based on FACS sorting threshold criteria. How would the analysis change if the dataset represented natural proportions of cell types? Presenting the data in this way can be confusing. Is there a way to present the data so that it becomes independent of how many cells per cell type are included? Would *Npy, Sst, Vip* for example have much lower peak FPKM and pFPKM percentile/rank? Similarly, Figure 1B and Table 3 (Fraction of pairs) do not represent biological distributions but are strongly influenced by enrichment of certain cell types. Table 3: would it be better to present fraction of cell type pairs rather than fraction of cell pairs for same reason?

2) Figure 4: Analysis of the transcriptomic data using the autoencoder was central to many of the main findings, but there was an overall lack of discussion of both the methodology and details of the features learned by the autoencoder. We have listed several areas for further discussion below:

How was the autoencoder architecture chosen, and why is there an increase rather than a decrease in the number of dimensions for the NP autoencoder (47 to 50 vs. 6,083 to 100)? Were there other architectures tested that did not perform as well as the one presented in this study? Please discuss this. Related, it was unclear why the authors chose to use the HE gene set instead of the 4,000 differentially expressed (DE) genes for WGCNA in Tasic et al., 2018. Did using the HE genes perform better or worse than taking the most variable or differentially expressed genes? Also, for the sets of 47 random genes, the authors could have matched these "random" sets to the NP gene set by measures of variability or differential expression instead of matching expression levels. This would make for a more interesting comparison than the random 47-gene sets drawn from all genes in the Tasic et al., 2018, dataset, since many of these randomly drawn genes might not be differentially expressed.

3) What do the features in the 5-d latent space of the autoencoder networks look like in terms of gene weights, and how do these dimensions/vectors compare to the principal components? This will also potentially help with clarifying/understanding the nature of the input used by the GMM classifier for classifying the cells that may subsequently affect the resolution index.

4) Figures 6, 7, and supplements to Figure 6: The coupling scores and matrices in these figures were important for inferring or predicting neuropeptide signaling between neuronal cell types. However, the authors provided little discussion of any significant trends or differences beyond their presentation of the individual coupling matrices for single cognate pairs. The study will be more impactful if the authors could provide more detailed discussion regarding the structure within these coupling matrices, with examples of specific differences or generalized trends observed from that structure to generate specific experimental hypotheses. Some specific points for further discussion are listed below:

Why was the threshold set to 50th percentile? Why not at certain CPM? The reasoning and method here is not completely clear. It seems that this strict threshold produces too many false negatives that may unnecessarily discard biologically plausible interactions to be evaluated.

The coupling matrices are all presented at the cell type level and not at the subclass or "family" level (branches/clades between subclass and type), although there is clearly some clustering or block structure at these higher levels along the hierarchical tree. Is NP signaling more type-specific or generalizable to a family/subclass? Perhaps a heatmap visualization at varying levels besides cell types/leaves more similar to Figure 5 will help.

5) The authors could provide more details on which cognate pairs are region-specific vs. "conserved" across regions based on the sparsity of the corresponding coupling matrix, and to compare this to the known properties of the corresponding NP-expressing cell type/family/subclass (e.g. the disinhibitory action of *Vip*interneurons via inhibition of *Sst*/Pvalb interneurons). Do the 47 pairs distribute in a laminar specific pattern and does this perhaps suggest whether they are used to augment bottom up, top-down or recurrent cortical activity? Also, the authors could comment more on the density/sparsity for each NP. Are there NPs that seem to have very specific signals (almost one-to-one), and which ones are much broader? For example, Trh->Trhr appears sparse/type-specific, whereas Adcyap1 appears to have a broad range of targets via multiple receptors and *Crh*->Crhr1/2 appears to act at an intermediate scale. Please provide more discussion of the variation in the sparsity of coupling for the cognate pairs analyzed.

6) Related to point 5, could the authors also provide more discussion of autocrine vs. paracrine NP signaling since there seem to be low coupling scores along the diagonal. This may suggest that there are very specific NPs that may act specifically in an autocrine manner, perhaps for autoinhibition in the case of Gi/o coupled pathways.

7) The presentation of the coupling matrices for single cognate pairs makes it difficult to appreciate structure/trends by the predominant coupling of downstream pathways. Are there preferred/predominant couplings for each subclass/family/type, e.g. mostly Gi-coupled for *Sst*interneurons?

8) In Figure 7, it would be helpful if the authors could provide an example of a network graph diagram depicting their inferred NP signaling for a particular cell type or subclass, such as the VISp *Vip*interneurons.

---

## [Author Response]

Essential revisions:1) Table 1: Since the dataset is enriched for certain cell types including Vip, Sst and Ndnf (partly Npy) cells, it is not surprising that these peptides appear high up in the list. The FACS sorting process probably even selects cells with higher peptide expression within these subtypes based on FACS sorting threshold criteria. How would the analysis change if the dataset represented natural proportions of cell types? Presenting the data in this way can be confusing. Is there a way to present the data so that it becomes independent of how many cells per cell type are included? Would Npy, Sst, Vip for example have much lower peak FPKM and pFPKM percentile/rank? Similarly, Figure 1B and Table 3 (Fraction of pairs) do not represent biological distributions but are strongly influenced by enrichment of certain cell types. Table 3: would it be better to present fraction of cell type pairs rather than fraction of cell pairs for same reason?

This query raises (a) the possibility that our peak FPKM values might be artifactually elevated by the recombinase-driven, FACS-based cell sampling process used by Tasic et al., 2018 to enrich collection of rare cell types, and (b) the likelihood that our results would be more forceful if expressed in terms of natural cell-type proportions, rather than proportions skewed by FACS-based type enrichment.

a) Are peak FPKM estimates artifacts of cell collection biases? As we have now attempted to describe more fully in our new Materials and methods section, our peak FPKM metric was designed to be as insensitive as possible to the absolute number of cells surveyed. The reviewers are nonetheless correct to note that it is conceivable that numbers could be influenced somehow by differential recombinase expression and FACS sorting. Such skewing would seem most likely to arise where sorting was keyed by xFP expression driven by an NPP-gene-based recombinase line (9 out of the 55 used by Tasic et al., 2018, see their Extended Data Figure 8). To address this possibility, we have compared peak FPKM scores extracted across the entire Tasic dataset (as reported in our Table 1 and below as “Tasic 2018 Mixture”) with pFPKM values extracted from four of their most heavily represented recombinase-driven subsets (with actual sorted cell counts): Gad2-IRES-Cre (2131), Slc32a1-IRES-Cre (2306), Vip-IRES-Cre (569), Sst-IRES-Cre (551). Of these four subsets, two (Vip-IRES-Cre and Sst-IRES-Cre) are driven by NPP-based recombinase lines and two by non-NPP lines (GAD2 and Slc32a1). The results are tabulated below:

Peak FPKM across Mixed and Cre-Restricted Neuron PopulationsRankGeneTasic 2018 MixtureGad2- IRES-CreSlc32a1-IRES-CreVip- IRES-CreSst- IRES-Cre1*Npy*108,865113,90493,70356,104103,1882*Sst*70,27474,99966,10013567,0953*Vip*48,74756,20148,89933,217294*Tac2*18,28417,75218,38021,2125,3956*Cck*16,39617,58618,80428,7111,5808*Penk*11,16012,22714,0989,7193,70510*Crh*9,11810,7359,1188,6052,05815*Cort*7,4777,7796,3164,1869,11018*Tac1*5,7285,4674,818748,65368*Pdyn*2,8133,1383,34303,276156*Pthlh*1,6561,5611,7991,969194509*Pnoc*698702758759385766*Trh*510895580968*Grp*435150101273291757*Rln1*258179160783032788*Adcyap1*1654964003917*Nts*12161162175324270*Nmb*11246343916

This table shows the variations expected from design of the NPP-based recombinase lines and prevailing knowledge of interneuron cell types (e.g., low expression of *Sst* in the *Vip* line and low expression of *Vip* in the *Sst* line), but also shows clearly that the high peak FPKM values we report in Table 1 are observed in the large GABA neuron subsets sorted by non-NPP-based recombinase expression. Based on these observations, we believe it is unlikely that the high NPP gene expression levels we report in Table 1 are artifact of the Tasic et al. FACS-sorted single-cell harvesting regime.

b) Does cell collection bias constrain interpretation of results? We concur wholeheartedly that our results could be more forceful and useful if the Tasic data and our analysis accurately reflected natural proportions of cell types. Unfortunately, it was judged impractical at the time Tasic et al. collected their data to assure such quantification, for two reasons. First, it was anticipated based on previous literature that some interneuron cell types would be so rare as to require prohibitive effort and expense to reach rare types via unbiased cell collection. Second, it was apparent in pilot studies that substantial collection bias would be unavoidable due to differential retention of different cell types and their mRNA during the process of single cell isolation. We have added columns labels “#Cells” to Figure 3—figure supplement 1. While these numbers do not reflect natural cell-type proportions, they at least indicate the actual numbers of cells upon which the statistics of each Tasic et al. type are based. We anticipate that developments of highly multiplexed FISH and other spatial transcriptomics methods now under way at the Allen Institute and other institutions will soon enable the very desirable ability to link transcriptomic neuron types to natural type proportions and many other essentially anatomic characteristics.

We have added one new analysis as a check for possible artifacts of neuron-type-dependent collection bias. We tested for a possible dependence of autoencoder clustering results, as illustrated in Figure 4, on neuron type. The result, shown in the new Figure 4—figure supplement 1, is that clustering shows little or no type-dependence, adding some credence to our original interpretation of the main autoencoder results.

c) Would it be better to present fraction of cell type pairs rather than fraction of cell pairs (in Table 3) for same reason? Thank you for the suggestion! We agree that presenting “fraction of type pairs” will be more informative and have modified Table 3 accordingly.

2) Figure 4: Analysis of the transcriptomic data using the autoencoder was central to many of the main findings, but there was an overall lack of discussion of both the methodology and details of the features learned by the autoencoder. We have listed several areas for further discussion below:How was the autoencoder architecture chosen, and why is there an increase rather than a decrease in the number of dimensions for the NP autoencoder (47 to 50 vs. 6,083 to 100)? Were there other architectures tested that did not perform as well as the one presented in this study? Please discuss this. Related, it was unclear why the authors chose to use the HE gene set instead of the 4,000 differentially expressed (DE) genes for WGCNA in Tasic et al., 2018. Did using the HE genes perform better or worse than taking the most variable or differentially expressed genes? Also, for the sets of 47 random genes, the authors could have matched these "random" sets to the NP gene set by measures of variability or differential expression instead of matching expression levels. This would make for a more interesting comparison than the random 47-gene sets drawn from all genes in the Tasic et al., 2018, dataset, since many of these randomly drawn genes might not be differentially expressed.

We have expanded the description of our autoencoder analysis methodology substantially, both in Results and in our new Materials and methods section and explained our architecture choice in more detail. We have also conducted additional experiments with architecture hyperparameters as requested. Briefly, very similar results were obtained for different choices of the autoencoder architecture. That is, the qualitative results depend only weakly on the hyperparameters of the network architecture.

Both before our original submission and after receiving this review, we have carried out much additional autoencoder analysis pertinent to these queries. We have now added the most germane of these results to our presentation, particularly in Figure 4D and related narrative. As requested, we now compare clustering performance based on the Tasic, et al., 2018, 4,020 DE gene set with that based on the 6,083 HE genes. As now illustrated by Figure 4D and described in text, these two large gene sets show very high and essentially indistinguishable performance close to the theoretical maximum possible. We also now report and discuss comparison of clustering performance based on the 47 NP genes and 47 DE genes that were selected as the most differentially expressed between Tasic 2018 neuron types. This comparison shows that the RI performance by the 47 DE gene set beats that of the 47 NP gene set by a small margin. We feel that our conclusion of exceptional clustering by 47 NP genes alone is still justified by the much better performance of the 47 NP set in comparison to every one of the 200 random 47-gene sets, and the fact that the 47 DE genes had the “unfair” advantage of being chosen with benefit of prior knowledge of the Tasic 2018 taxonomy.

3) What do the features in the 5-d latent space of the autoencoder networks look like in terms of gene weights, and how do these dimensions/vectors compare to the principal components? This will also potentially help with clarifying/understanding the nature of the input used by the GMM classifier for classifying the cells that may subsequently affect the resolution index.

The latent space representations are obtained by transforming the transcript expression data into a low-dimensional, abstract space. Individual dimensions of this space represent nonlinear combinations of gene expression and are not necessarily tied to a small subset of genes. Moreover, it would be incorrect to explain the latent space with simple linear weights. Indeed, the network is able to compress the data into so few dimensions by learning the (non-linear) inter-dependent nature of gene expression. While the underlying mechanics are quite different, the abstract nature of the representation of our autoencoder network is the same as that of the perhaps more familiar tSNE transform.

When one studies the expression of a particular gene in this space, the landscape would appear complex for many genes, essentially capturing the fact that the expression of single genes is rarely localized to single cell types. On the other hand, the markers of well-known cell classes such as *Vip, Sst*, and Pvalb, are indeed enriched in the “islands” corresponding to those cell classes, directly reflecting the ability to classify them correctly.

The reviewers draw a very good analogy with PCA. Indeed, the objective function used for the autoencoder network is the same as that of the well-known PCA, namely the mean-squared error. The difference is that the transformation from the high dimensional gene expression space to the low-dimensional representation space is restricted to be linear in the case of PCA. Applying PCA to single-cell RNA sequencing data often produces only three ‘blobs’ corresponding to the excitatory, inhibitory, and non-neuronal cell classes. The finer distinctions within those classes are, however, lost because PCA fails to identify further structure within those ‘blobs’.

4) Figures 6, 7 and supplements to Figure 6: The coupling scores and matrices in these figures were important for inferring or predicting neuropeptide signaling between neuronal cell types. However, the authors provided little discussion of any significant trends or differences beyond their presentation of the individual coupling matrices for single cognate pairs. The study will be more impactful if the authors could provide more detailed discussion regarding the structure within these coupling matrices, with examples of specific differences or generalized trends observed from that structure to generate specific experimental hypotheses. Some specific points for further discussion are listed below:Why was the threshold set to 50th percentile? Why not at certain CPM? The reasoning and method here is not completely clear. It seems that this strict threshold produces too many false negatives that may unnecessarily discard biologically plausible interactions to be evaluated.The coupling matrices are all presented at the cell type level and not at the subclass or "family" level (branches/clades between subclass and type), although there is clearly some clustering or block structure at these higher levels along the hierarchical tree. Is NP signaling more type-specific or generalizable to a family/subclass? Perhaps a heatmap visualization at varying levels besides cell types/leaves more similar to Figure 5 will help.

What trends or differences do the coupling matrices reveal? We see now that we may have been too limited in our qualitative and quantitative description of the widely varied structures of our peptidergic network coupling predictions. While we are indeed fascinated by these structures, we have been somewhat reticent in elaborating upon them, because they are still only predictions. Much neuroanatomy, cell biology, physiology and biophysics remains to be done to test these predictions and cast them empirically then as facts (or fancy). We feel that our text as it stands discusses these reservations and their implications reasonably well. Having said that, we have embraced the spirit of this query enthusiastically with the substantial increments to network matrix characterization we discuss below and in connection with the next query #5.

How is the coupling matrix scheme justified? This suggestion/query inspired us to redesign and simplify our schemes for predicting and presenting coupling adjacency matrices. We now predict coupling as a straight outer product with NPP and NP-GPCR gene expression by cell type in CPM as of row and column vectors, deleting the former thresholding step (and with it the troublesome need to set a threshold value; mitigating also the tendency toward excess false negatives). Logarithmic color-scale compression is used to accommodate display of a very wide dynamic range of CPM*CPM signals. This scheme is laid out in graphical detail in our revised Figure 6A and applied to all additional examples in Figure 6 and Figure 6—figure supplement 1. As before, we also provide the underlying data and code to regenerate any or all matrices. We are grateful for the reviewer comment that led to this simplifying improvement.

How does peptidergic coupling look at higher taxonomic levels (e.g., subclasses rather than leaves)? Here we have augmented our presentation here very substantially. In Figures 3, 5 and 6 (and the new Figure 7), we have overlaid graphic color block fills and guidelines which demarcate subclass levels of the Tasic 2018 neurotaxonomy on our type-level expression heatmaps and coupling matrix plots. These graphics are intended to make it much easier to visualize patterns at higher taxonomic levels while still revealing variations at finer levels. We have also added a new analysis (depicted in new panels Figures 3C and 3D) to quantify the residual type-dependent gene expression variance after pooling at the level of subclasses. Additionally, a new panel (6F) added to Figure 6 and new panels for every cognate pair added to Figure 6—figure supplement 1 show coupling matrix prediction pooled at the subclass level establish that differential structures of predicted peptidergic coupling persists strongly at higher levels of the Tasic 2018 hierarchy, while at the same time revealing that strong and conserved variations persist at the level of types/leaves.

5) The authors could provide more details on which cognate pairs are region-specific vs. "conserved" across regions based on the sparsity of the corresponding coupling matrix, and to compare this to the known properties of the corresponding NP-expressing cell type/family/subclass (e.g. the disinhibitory action of Vip interneurons via inhibition of Sst/Pvalb interneurons). Do the 47 pairs distribute in a laminar specific pattern and does this perhaps suggest whether they are used to augment bottom up, top-down or recurrent cortical activity? Also, the authors could comment more on the density/sparsity for each NP. Are there NPs that seem to have very specific signals (almost one-to-one), and which ones are much broader? For example, Trh->Trhr appears sparse/type-specific, whereas Adcyap1 appears to have a broad range of targets via multiple receptors and Crh->Crhr1/2 appears to act at an intermediate scale. Please provide more discussion of the variation in the sparsity of coupling for the cognate pairs analyzed.

Are cognate pair distributions laminar specific? A preliminary analysis reveals only minor laminar specificity of cognate pair distributions. The Tasic 2018 data (their Figure 5) shows that the most marked laminar type/subclass differential for interneurons is that distinguishing *Sst* vs. *Vip*subclasses. In Author response image 1, we add yet another color code and a set of matrix overlay lines to indicate upper vs. lower layer cell body harvest origins predominating by neuron type for representative *Sst* and *Vip* cognate pairs.

These plots do not reveal much layer specificity within either subclass. This result has discouraged us from digging much more deeply into the presently available data. We feel we may be at the limits of the microanatomical information we can extract within the technical confines of the present data source. Furthermore, we are aware of spatial transcriptomic data likely to emerge soon from various research sites that should allow for much more conclusive investigation of the important issues raised here by the reviewers. We’d prefer to avoid overloading this report with more analysis along these lines when greatly improved clarity from new experimental findings seems imminent.

How does the density/sparsity of predicted type-coupling vary with cognate pair identity? These queries have inspired us to develop quantitative measures of coupling matrix structure to allow for comparisons across all cognate pairs and both cortical regions. New results linked to Figure 6 (figure supplements 2 and 3) delineate strong variations in sparsity and specificity amongst the 37 cognate pairs and confirm strong conservation between VISp and ALM areas, while also establishing that redundancy of overall architectures of the predicted 37 networks is rather minimal. As requested, we have also included more discussion on these topics.

6) Related to point 5, could the authors also provide more discussion of autocrine vs. paracrine NP signaling since there seem to be low coupling scores along the diagonal. This may suggest that there are very specific NPs that may act specifically in an autocrine manner, perhaps for autoinhibition in the case of Gi/o coupled pathways.

Indeed, there are fascinating indications of possible autocrine signaling in co-expression of cognate pairs evident in the Tasic 2018 data. We have linked a new Supplementary figure to Figure 2 to provide a preliminary glimpse into the structure of possible autocrine peptidergic signaling in mouse neocortex. Interestingly, the more strongly co-expressed autocrine pairs represent all three of the primary Gα families we consider, although there is a notable preponderance of Gi/o.

We are very interested in further exploration of possible autocrine neuropeptide signaling, and plan to actively pursue a complete analysis including experimental work, but we have felt that inclusion of this tangent here would risk serious overloading an already very lengthy report without doing justice to an important topic for future transcriptomic and physiological research.

7) The presentation of the coupling matrices for single cognate pairs makes it difficult to appreciate structure/trends by the predominant coupling of downstream pathways. Are there preferred/predominant couplings for each subclass/family/type, e.g. mostly Gi-coupled for Sst interneurons?

Motivated by the very premise put forth in this query, we have explored many approaches to compact visualization of predominant downstream receptor coupling modes for 37 cognate pairs. We are pleased to offer a new figure (Figure 7) and accompanying narrative that we feel begins to address this issue fairly effectively and may help guide future efforts at empirical tests of our coupling predictions.

8) In Figure 7, it would be helpful if the authors could provide an example of a network graph diagram depicting their inferred NP signaling for a particular cell type or subclass, such as the VISp Vip interneurons.

The network integration “cartoon” schematic (formerly Figure 7, now Figure 8) has been redrawn and re-framed to convey more clearly our original intent, which was to summarize our basic methodology while highlighting the likely future importance of neurotaxonomy as means to integrate statistical descriptions of multiple neuronal signaling networks, be they modulatory, synaptic or both. We apologize if this message seems a bit orthogonal to the more concrete and specific direction the reviewers suggest here. We hope that we have now stated our intent more clearly and that the reviewers find this dose of speculation reasonably justifiable and worthwhile here.